# DTP:A Simple yet Effective Distracting Token Pruning Framework for Vision-Language Action Models

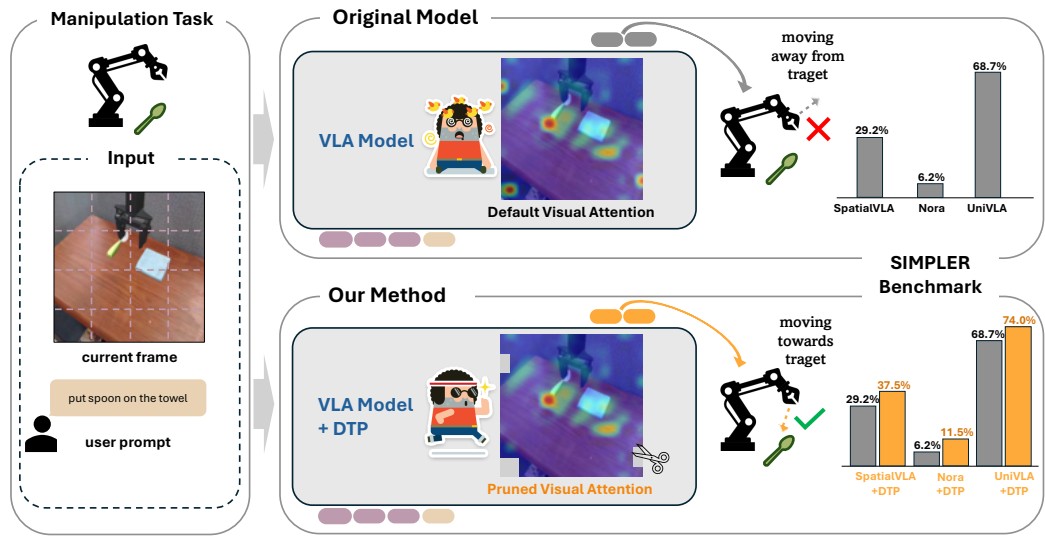

Figure 1: **Overview of our proposed Distracting Token Pruning (DTP) method for improving Vision-Language-Action (VLA) models in robotic manipulation.** Left: Input consisting of the current visual observation and natural language instruction. Middle: Comparison between the original VLA model (top) may focus on task-irrelevant regions that lead to task failure, while our DTP-enhanced approach (bottom) creates more focused attention on task-critical areas, leading to the improvements in the task success rate.

## ABSTRACT

Vision-Language Action (VLA) models have shown remarkable progress in robotic manipulation by leveraging the powerful perception abilities of Vision-Language Models (VLMs) to understand environments and directly output actions. However, by default, VLA models may overly attend to image tokens in the task-irrelevant region, which we describe as 'distracting tokens'. This behavior can disturb the model from the generation of the desired action tokens in each step, affecting the success rate of tasks. In this paper, we introduce a simple yet effective plug-and-play **D**istracting **T**oken **P**runing (DTP) framework, which dynamically detects and prunes these distracting image tokens. By correcting the model's visual attention patterns, we aim to improve the task success rate, as well as exploring the performance upper boundaries of the model without altering its original architecture or adding additional inputs. Experiments on the SIMPLER Benchmark (Li et al., 2024) show that our method consistently achieving relative improvements in task success rates across different types of novel VLA models, demonstrating generalizability to transformer-based VLAs. Further analysis reveals a negative correlation between the task success rate and the amount of attentions in the task-irrelevant region for all models tested, highlighting a common phenomenon of VLA models that could guide future research. We also publish our code at: https://anonymous.4open.science/r/CBD3.

## 1 INTRODUCTION

The success of Vision-Language Models (VLMs) in understanding and reasoning about visual content has opened new possibilities for embodied AI. Trained on large-scale image–text data, VLMs (Liu et al., 2023b; Karamcheti et al., 2024; Beyer et al., 2024; Lu et al., 2024; Malartic et al., 2024; Chen et al., 2025; Bai et al., 2025) excel at visual recognition, reasoning, and visual question answering. Building on this, Vision-Language-Action (VLA) models extend VLMs to generate executable robot actions, bridging high-level semantic reasoning with low-level control. Early VLAs such as RT-1 (Brohan et al., 2023b) and RT-2 (Brohan et al., 2023a) pioneered action tokenization for robotic control, while later models like OpenVLA (Kim et al., 2024) achieved strong performance despite smaller sizes. These advances established the core VLA pipeline: a visual encoder (Radford et al., 2021; Sun et al., 2023; Zhai et al., 2023; Oquab et al., 2024), a language model (Touvron et al., 2023; Dai et al., 2024; Qwen et al., 2025), and discrete action token generation.

Recent works further improve VLA capabilities. SpatialVLA (Qu et al., 2025) and (Sun et al., 2025; Lin et al., 2025; Li et al., 2025) incorporate depth for better 3D understanding, Nora (Hung et al., 2025) employs tokenizers such as FAST (Pertsch et al., 2025) for precise continuous control, and world-model approaches (Cen et al., 2025; Wang et al., 2025) leverage dynamics modeling for action prediction. Despite these architectural advances, VLAs may still fail in manipulation tasks due to redundant visual tokens, with task-irrelevant visual tokens may receive disproportionately high attention during cross-attention, distracting the model from task-relevant regions (Figure 1). This raises a concern: *Do distracting tokens affect task success, and can pruning them improve performance?*

We initially tried manually inspecting visual attention patterns and pruning image tokens for the model. However, this approach has two major limitations: (1) reviewing attention by human for each action token is extremely time-consuming, especially since the model can output hundreds to thousands action tokens for each episode, making full test-suite analysis impractical; (2) human intuition about "good" attention patterns may not align with the model's preferences. Although ideal attention should focus on task-relevant objects and regions, different VLA models may have their own visual pattern preference, so manual pruning does not necessarily lead to improved performance.

To address these two problems, we propose **Distracting Token Pruning (DTP)**, a plug-and-play framework that dynamically prunes distracting tokens. DTP contains three stages: (1) Relevance-based important region construction, where prompt–image token interactions identify task-relevant visual tokens, and (2) Action attention analysis, where the attention heatmap reveals which image tokens the model focuses at each action generation step. (3) We then apply an intersection-based strategy, selectively pruning visual tokens in the unimportant region, if their attention exceeds the maximum attention within the important region (scaled by tolerance $\tau$). Experiments on the SIMPLER Benchmark (Li et al., 2024) prove that DTP effectively corrects visual attention patterns, improves task success rates, and generalizes across transformer-based VLAs.

In summary, our work makes the following contributions: (1) We introduce the Distracting Token Pruning (DTP) framework, a novel intersection-based method that automatically identifies and prunes distracting tokens, improving task success rates by addressing common attention weaknesses in VLA models. (2) We explore the performance upper bound of VLAs by varying tolerance $\tau$, seeking the ideal visual attention patterns that align with model preferences and maximize achievable performance. (3) We analyze attention values in unimportant regions, revealing a negative correlation with task success and offering insights for building more robust VLAs.

## 2 RELATED WORK

**Vision-Language Models (VLMs)** have become the foundation for multimodal reasoning, showing strong generalization across visual understanding, captioning, and visual question answering. Large-scale pretraining on internet-scale image–text data has enabled models such as PrismaticVLM Karamcheti et al. (2024), InternVL2.5 Chen et al. (2025), PaliGemma 2 Beyer et al. (2024), and Qwen2.5-VL Bai et al. (2025) to achieve impressive performance on various visual-language tasks. These models combine high-capacity visual encoders such as SigLIP Zhai et al. (2023) and DINOv2 Oquab et al. (2024) with large language models (Touvron et al., 2023; Dai et al., 2024;

Qwen et al., 2025), producing flexible representations that can be adapted across a wide range of tasks. While primarily designed for static vision-language tasks, the success of VLMs in structured reasoning has inspired their extension into embodied AI domains. In particular, their ability to jointly ground natural language with perceptual input provides a natural interface for robotic control. This transition, however, requires not only high-level semantic understanding but also the capacity to map instructions and observations into temporally grounded motor actions — a gap that gave rise to Vision-Language-Action (VLA) models.

**Vision-Language-Action Models (VLAs)** extend the capabilities of VLMs by directly generating robot actions, effectively bridging perception and motor control. Early work such as RT-1 (Brohan et al., 2023b) and RT-2 (Brohan et al., 2023a) established the paradigm of representing actions as discrete tokens, enabling pretrained language models to output robot commands in a manner analogous to text generation. Building on this foundation, OpenVLA (Kim et al., 2024) and OpenVLA-OFT (Kim et al., 2025) further reduced the model size and improved inference efficiency. Recent efforts have pushed VLAs in two complementary directions. First, improvements in vision encoding aim to overcome limitations of purely 2D perception. SpatialVLA (Qu et al., 2025), GeoVLA (Sun et al., 2025), EVO-0 (Lin et al., 2025), and PointVLA (Li et al., 2025) incorporate the depth information for the objects in the input image, boosting spatial perception ability and yielding the increase in task success rate, Second, advances in action decoding focus on bridging the gap between discrete action tokens and continuous control. Models such as Nora (Hung et al., 2025), $\pi_{0.5}$ (Intelligence et al., 2025), and $\pi_0$-FAST (Pertsch et al., 2025) transform discrete token outputs into continuous action values, enabling more precise and faster execution. Beyond architectural refinements, new paradigms such as UniVLA (Wang et al., 2025) and WorldVLA (Cen et al., 2025) integrate world-modeling into the VLA framework. By jointly predicting future observations and generating actions, world models capture the underlying physics of the environment, which helps for more accurate action token generation. However, existing VLAs still face challenges where models may attend heavily to task-irrelevant tokens, a phenomenon that can degrade action quality and lower task success rates. Several works have attempted to mitigate visual noise or highlight task-relevant regions. For example, BYOVLA (Hancock et al., 2024) proposes a run-time observation intervention scheme that modifies the input image based on visual sensitivity, while Otter (Huang et al., 2025) introduces an architecture that learns text-aware feature filtering through additional model components and end-to-end training. While these approaches improve robustness, they operate either at the input level or require architectural redesign and training, making them less broadly applicable to existing VLA systems. This limitation motivates our work: developing inference-time, training-free, and architecture-agnostic methods to detect and prune these distracting tokens, thereby refining the model's visual attention and improving the precision of action generation.

## 3 METHOD

Our method involves three stages to find and prune distracting tokens (see Figure 2). First, we analyze prompt-to-visual relevance to identify visual tokens that mostly related to the user's intent task. Next, we conduct a weighted self-attention analysis from the generation perspective. Finally, we apply a dynamic intersection-based pruning strategy that reconciles both perspectives, selectively pruning image tokens. This multi-stage designs help the model to automatically correct its visual attention patterns at each generation step.

### 3.1 PROBLEM FORMULATION

Given a VLA model $\mathcal{F}$ that processes visual tokens $\mathbf{V} \in \mathbb{R}^{M \times d}$ where $M$ represents number of image tokens, our objective is to identify a subset $\mathbf{V}' \subset \mathbf{V}$ with $|\mathbf{V}'| = K$ (where $K < M$) that prune the distracting image tokens, yielding improved task success rate $\text{Succ}(\mathcal{F}(\mathbf{V}')) \geq \text{Succ}(\mathcal{F}(\mathbf{V}))$.

#### 3.1.1 IMPORTANT REGION CONSTRUCTION

We compute the relevance score for visual tokens based on their interactions with prompt tokens. Let $\mathbf{P}$ denote the set of prompt tokens, $\mathbf{V}$ the set of visual tokens, and $\mathcal{C}$ the set of selected layers. For each layer $c \in \mathcal{C}$ and each prompt token $p_i \in \mathbf{P}$, we calculate the relevance scores to all visual

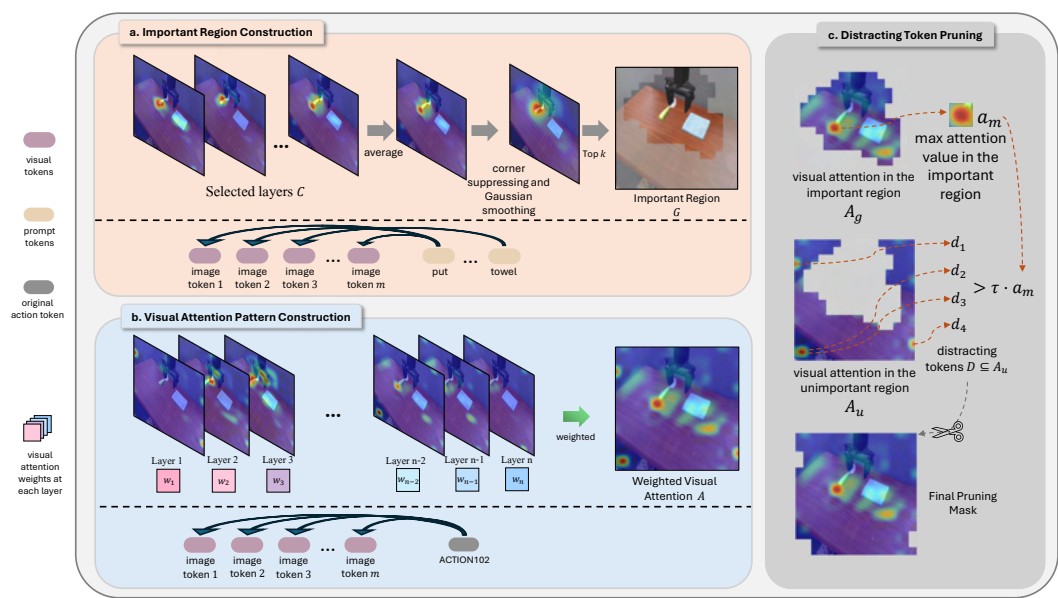

Figure 2: **Detailed architecture of our Distracting Token Pruning (DTP) method for improving visual attention in Vision-Language-Action models**. The method consists of three main components: **(a) Important Region Construction**: Using selected transformer layers $C$ to calculate the relevance score between image and prompt tokens, which forms the task-related important region $G$. **(b) Visual Attention Pattern Construction**: Creating the output to image token attention heatmap $A$ from all attention layers, weighted by the visual attention proportion. It captures where the model focuses when generating actions. **(c) Distracting Token Pruning**: For any image token in the unimportant region, if its attention value is greater than $\tau \cdot a_m$, it will be treated as distracting tokens, and will be pruned.

tokens as:

$$\mathbf{r}_{p_i}^c = \frac{1}{H} \sum_{h=1}^{H} \mathbf{A}_h^c[\, p_i, \mathbf{V} \,], \tag{1}$$

where $\mathbf{A}_h^c \in \mathbb{R}^{S \times S}$ is the attention matrix of head $h$ at layer $c$, and $S$ is the total number of tokens (text, visual, system, etc.). $\mathbf{A}_h^c[p_i, \mathbf{V}]$ selects the attention weights from prompt token $p_i$ to all visual tokens $\mathbf{V}$ in that head. Averaging over $H$ heads produces the relevance vector $\mathbf{r}_{p_i}^c \in \mathbb{R}^M$, representing the overall importance of each visual token to prompt token $p_i$ at layer $c$.

For models (such as UniVLA (Wang et al., 2025)), where the input sequence places image tokens *after* the prompt tokens, and prompt tokens don't have attention to image tokens, we instead compute the relevance heatmap using embedding similarity.

$$\mathbf{r}_{p_i}^c = \text{cosine}\big(\mathbf{E}_{p_i}, \mathbf{E}_{\mathbf{V}}\big), \tag{2}$$

where $\mathbf{E}_{p_i} \in \mathbb{R}^d$ and $\mathbf{E}_{\mathbf{V}} \in \mathbb{R}^{M \times d}$ are the embeddings of the prompt token and visual tokens, respectively. Then we can aggregate across both prompt tokens and layers in a single step, and obtain the overall relevance heatmap $R$ for visual tokens:

$$\mathbf{R} = \frac{1}{|\mathcal{C}|} \sum_{c \in \mathcal{C}} \frac{1}{|\mathbf{P}|} \sum_{i=1}^{|\mathbf{P}|} \mathbf{r}_{p_i}^c, \tag{3}$$

To reduce corner artifacts and focus on central regions, we apply spatial biasing with corner suppression and Gaussian smoothing. Finally, we identify the $k$ highest relevance visual tokens to form the important region $G$.

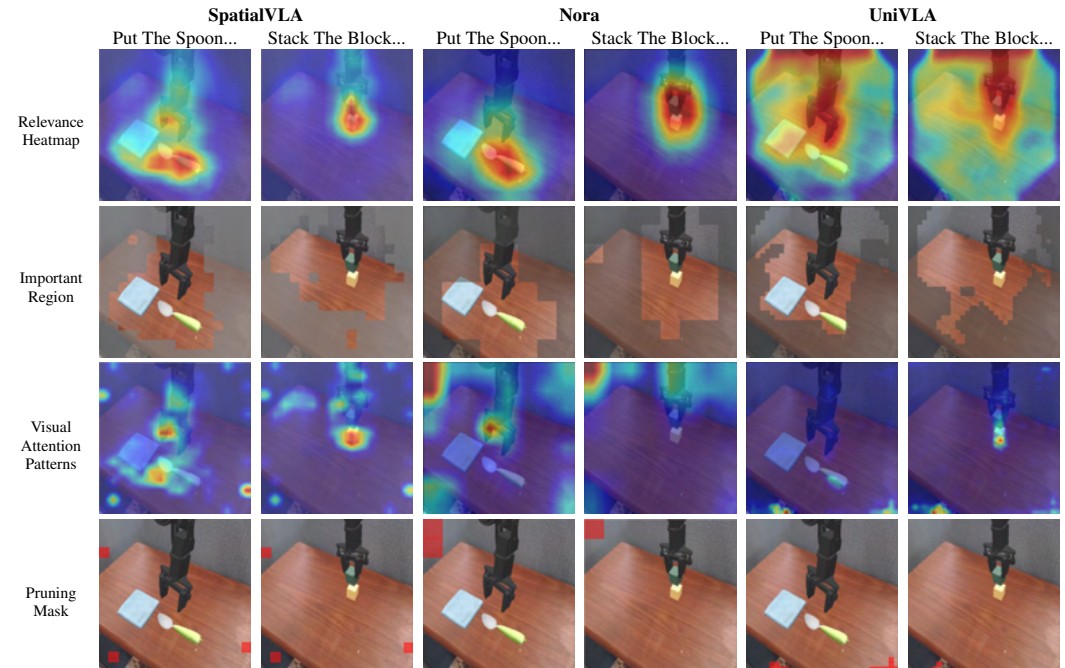

Figure 3: **Visualization of the distracting token pruning process across different VLA models and tasks.** The figure shows the Relevance Heatmap (first row), Important Region (second row), Visual Attention Patterns (third row), and Final Pruning Mask (last row) for three VLA models (SpatialVLA (Qu et al., 2025), Nora (Hung et al., 2025), UniVLA (Wang et al., 2025)) on two representative tasks ('Put The Spoon...' and 'Stack The Block...'). The comparison reveals how our method identifies and prunes distracting visual tokens across different model architectures. Please refer to Appendix A.4 for more visualization cases.

## 3.2 VISUAL ATTENTION PATTERN CONSTRUCTION

To analyze which visual tokens the model attends to during action generation, we compute a visual attention pattern for each generated action token. For each action token $t_j$ and each layer $l \in \{1, \ldots, N\}$, where $N$ is the total number of layers in the model. We first extract the attention weights from $t_j$ to all visual tokens, producing a layer-specific heatmap $\mathbf{A}_j^l \in \mathbb{R}^M$. We then weight each layer's heatmap by the proportion of total visual attention in that layer, denoted as $w^l$, and aggregate across layers to obtain the final visual attention pattern:

$$\mathbf{A}_j = \sum_{l=1}^{\mathbf{N}} w^l \mathbf{A}_j^l,\tag{4}$$

where $\mathbf{A}_j \in \mathbb{R}^M$ represents the overall attention the model assigns to each visual token while generating action token $t_j$. This final heatmap indicates which image tokens that model pay most of attentions during the action token generation.

## 3.3 DISTRACTING TOKEN PRUNING

Given the important region $\mathbf{G}$ obtained from the relevance heatmap and the visual attention pattern $\mathbf{A}$, we identify distracting visual tokens by comparing attention values inside and outside of $\mathbf{G}$.

Let $a_m$ denote the maximum attention value among visual tokens located within the important attention region $A_g$. For each visual token $v$ located in the unimportant attention region $A_u$, we denote it as a distracting token $d$ if its attention exceeds the thresholded maximum:

$$d \in \mathcal{D} \quad \text{iff} \quad A_u[v] > \tau \cdot a_m,\tag{5}$$

Table 1: **Evaluation across different policies on WidowX Robot tasks**. Our DTP method demonstrates significant improvements across complex manipulation tasks.

| Model | Put Spoon on Towel | | Put Carrot on Plate | | Stack Green Block on Yellow Block | | Put Eggplant in Yellow Basket | | SR | Rel. |
|---|---|---|---|---|---|---|---|---|---|---|
| | Grasp | Success | Grasp | Success | Grasp | Success | Grasp | Success | | |
| SpatialVLA (Qu et al., 2025) | 12.5% | 12.5% | 37.5% | 20.8% | 70.8% | 25.0% | 75.0% | 58.3% | 29.2% | 100% |
| SpatialVLA + DTP (Ours) | 25.0% | 25.0% | 45.8% | 33.3% | 54.2% | 29.2% | 62.5% | 62.5% | **37.5%** | **128.4%** |
| Nora (Hung et al., 2025) | 37.5% | 16.7% | 41.7% | 4.2% | 50.0% | 4.2% | 0.0% | 0.0% | 6.2% | 100% |
| Nora + DTP (Ours) | 37.5% | 20.8% | 41.7% | 4.2% | 45.8% | 12.5% | 8.3% | 8.3% | **11.5%** | **185.5%** |
| UniVLA (Wang et al., 2025) | 83.3% | 70.8% | 75.0% | 70.8% | 79.2% | 33.3% | 100.0% | 100.0% | 68.7% | 100% |
| UniVLA + DTP (Ours) | 87.5% | 75.0% | 79.2% | 75.0% | 100.0% | 45.8% | 100.0% | 100.0% | **74.0%** | **107.7%** |

Table 2: **Evaluation across different policies on Google Robot tasks**. Our DTP method shows consistent improvements across different VLA architectures.

| Model | Visual Matching | | | | Variant Aggregation | | | |
|---|---|---|---|---|---|---|---|---|
| | Pick Coke Can | Move Near | SR | Rel. | Pick Coke Can | Move Near | SR | Rel. |
| SpatialVLA (Qu et al., 2025) | 83.3% | 75.4% | 79.4% | 100% | 90.2% | 68.7% | 79.5% | 100% |
| SpatialVLA + DTP (Ours) | 86.7% | 77.1% | **81.9%** | **103.1%** | 90.2% | 70.7% | **80.5%** | **101.3%** |
| Nora (Hung et al., 2025) | 53.7% | 43.8% | 48.7% | 100% | 56.0% | 44.7% | 50.4% | 100% |
| Nora + DTP (Ours) | 54.7% | 45.0% | **49.9%** | **102.4%** | 56.7% | 47.0% | **51.9%** | **103.0%** |

Table 3: **Evaluation of Nora on the LIBERO Benchmark**. Our DTP method consistently improves performance across all LIBERO suites, including the challenging LIBERO-10.

| Model | LIBERO Benchmark | | | |
|---|---|---|---|---|
| | LIBERO-Spatial | LIBERO-Object | LIBERO-Goal | LIBERO-10 |
| Nora (Hung et al., 2025) | 92.8% | 76.0% | 85.8% | 69.6% |
| Nora + DTP (Ours) | **93.0%** | **77.4%** | **88.4%** | **76.2%** |

Where $\tau$ is a tolerance factor and $\mathcal{D}$ is the set of all distracting tokens. Finally, all distracting tokens in $\mathcal{D}$ will be pruned from the input to generate the refined action token. Please refer to Figure 3 for the visualization of the process for different models and tasks.

# 4 EXPERIMENTS

We evaluate DTP on the SIMPLER Benchmark (Li et al., 2024) using three state-of-the-art VLAs: SpatialVLA (Qu et al., 2025), a 3B model based on Paligemma 2 (Beyer et al., 2024) with integrated 3D position encoding; Nora (Hung et al., 2025), built on Qwen2.5-VL-3B (Bai et al., 2025) with an additional FAST tokenizer (Pertsch et al., 2025) for action decoding; and UniVLA (Wang et al., 2025), a 9B world model for VLA tasks. This diverse set of models enables us to test DTP's generalizability across different architectures and demonstrate its effectiveness in improving robotic manipulation performance.

## 4.1 MAIN RESULTS

Our experiments across both SIMPLER Benchmark and LIBERO Benchmark (Liu et al., 2023a) demonstrate that DTP consistently improves task success rates for transformer-based VLAs. We report results using **SR** (average task success rate) and **Rel.** (relative success rate), where the latter measures improvement over the baseline model. For each VLA checkpoint (SpatialVLA, Nora, UniVLA), we choose one global set of hyperparameters: tolerance $\tau$, selected layers $C$ for relevance heatmaps, number of top-$k$ relevant tokens and Gaussian smoothing parameters $\sigma$. **This set of hyperparameters is kept fixed across all tasks, rollouts, and environments** for each model checkpoint. Which means we do not tune hyperparameters per task in this section.

On **WidowX tasks** (Table 1), SpatialVLA improves from 29.2% to 37.5% average success (+28.4% relative), Nora rises from 6.2% to 11.5% (nearly 2×), and UniVLA, already the strongest baseline (68.7%), further increases to 74.0% (+7.7% relative). These results indicate that DTP benefits both weak and strong policies by pruning distracting tokens.

On **Google Robot tasks** (Table 2), improvements are smaller but consistent. SpatialVLA increases by +1–3% relatively, while Nora gains +2–3% relatively, confirming that DTP generalizes across robots and architectures, including already strong-performing setups. (Since UniVLA does not provide checkpoints for the Google Robot embodiment tasks, we exclude it from this test suite.)

To further evaluate the generalization of DTP beyond SIMPLER Benchmark, we additionally test **Nora** on the **LIBERO Benchmark** (Liu et al., 2023a). We choose Nora because its official codebase and pretrained checkpoints for LIBERO are publicly available, enabling a reliable and reproducible evaluation. The results are shown in Table 3.

Our DTP method consistently improves Nora across all LIBERO suites. Notably, it yields a $+6.6\%$ absolute gain on the more challenging LIBERO-10, and $+1.4$–$2.6\%$ improvements on the other benchmarks, despite the already high performance of the base model. This demonstrates that the effectiveness of DTP generalizes beyond SIMPLER and extends to more diverse and complex manipulation benchmarks.

Overall, the results highlight two key findings: **Robustness across models**: DTP benefits both strong (UniVLA) and weak (Nora) VLAs, suggesting that pruning distracting visual tokens universally reduces noise in decision-making. **Generalizability across benchmarks, tasks and robots**: Improvements are observed across different environments and embodiments, indicating that DTP is not task-specific but broadly applicable. Detailed implementations are provided in Appendix A.2.

## 4.2 Performance Upper Bound under Visual Attention Patterns

To investigate the performance upper bound achievable by a VLA model under its *existing architecture*, we analyze how different visual attention patterns influence the model's uncertainty about the correct action token $A^*$ (see Appendix A.1 for assumptions).

Let $Z \in \mathbb{R}^{M \times d}$ denote the matrix of visual token value vectors, where $M$ is the number of visual tokens and $d$ is the feature dimension. Given a visual attention pattern $\alpha$, the model produces a modified set of value vectors $Z_\alpha$, reflecting which regions of the image the model focuses on. Changing $\alpha$ alters the visual representation and therefore the model's uncertainty about the correct action.

**Conditional uncertainty.** We define the uncertainty of the model under attention pattern $\alpha$ as

$$E(\alpha) = H(A^* \mid Z_\alpha), \tag{6}$$

which measures how uncertain the model remains about the correct action token after observing the visual features selected by $\alpha$. A high value of $E(\alpha)$ indicates that the attention pattern includes distracting or irrelevant tokens, while a low value indicates that the attention emphasizes task-relevant regions.

**Normalized performance score.** To measure how much useful information the attention pattern provides relative to the inherent ambiguity in the task, we normalize by the entropy of the correct action distribution:

$$P(\alpha) = 1 - \frac{E(\alpha)}{H(A^*)}, \tag{7}$$

where $H(A^*)$ captures the baseline uncertainty regarding the correct action before conditioning on visual input. Thus, $P(\alpha) \in [0, 1]$ serves as a normalized performance measure: $P(\alpha) = 1$ *denotes perfect certainty*, while $P(\alpha) = 0$ *indicates that the visual attention pattern provides no useful information*.

**Optimal attention pattern.** The optimal attention configuration $\alpha^*$ is defined as the pattern that minimizes the conditional uncertainty:

$$P(\alpha^*) = 1 - \frac{H(A^* \mid Z_{\alpha^*})}{H(A^*)}, \quad \alpha^* \in \arg\min_\alpha E(\alpha). \tag{8}$$

This represents the best achievable performance under the model's architecture and visual encoder, without modifying weights, inputs, or structure.

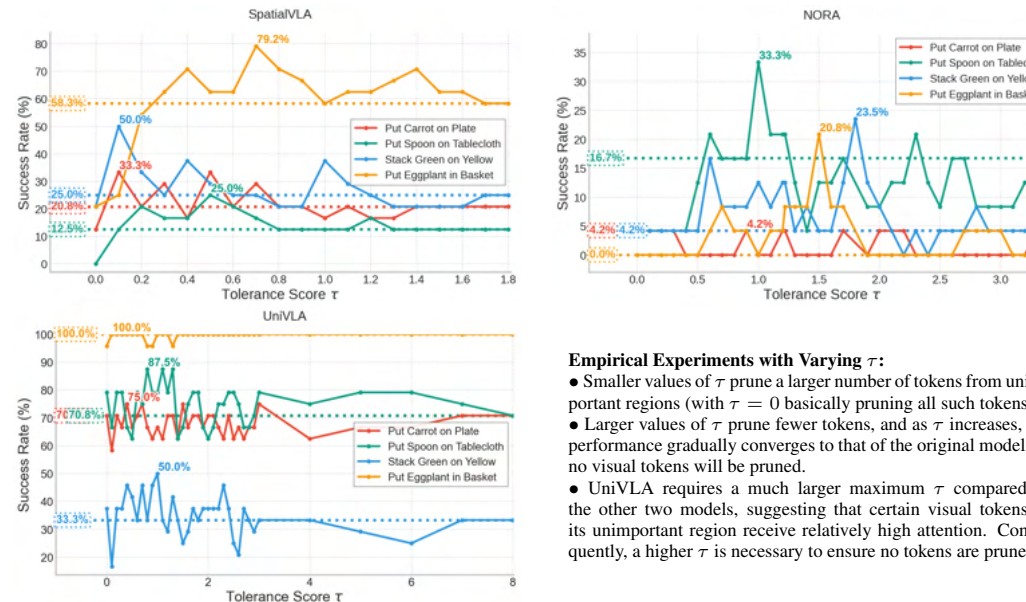

**Empirical Experiments with Varying $\tau$:**
• Smaller values of $\tau$ prune a larger number of tokens from unimportant regions (with $\tau = 0$ basically pruning all such tokens).
• Larger values of $\tau$ prune fewer tokens, and as $\tau$ increases, the performance gradually converges to that of the original model, as no visual tokens will be pruned.
• UniVLA requires a much larger maximum $\tau$ compared to the other two models, suggesting that certain visual tokens in its unimportant region receive relatively high attention. Consequently, a higher $\tau$ is necessary to ensure no tokens are pruned.

Figure 4: **Exploration of performance upper bounds under varying tolerance parameter $\tau$ across different VLA models**. Dashed lines denote the baseline success rates without DTP, while solid lines illustrate performance under DTP with different $\tau$ values. The success rate is annotated at peak to indicate $\hat{\tau}$. The results confirm that tuning $\tau$ enables the selection of optimal tolerance levels, thereby maximizing potential performance gains.

**Approximating the optimal pattern via DTP.** In practice, $\alpha^*$ is unknown. DTP provides an approximation by pruning distracting tokens based on a tolerance factor $\tau$, yielding an induced attention pattern $\alpha_\tau$. By sweeping over $\tau$, we identify $\hat{\tau}$ that maximizes $P(\alpha)$:

$$P(\alpha_{\text{def}}) \leq P(\hat{\alpha}) \leq P(\alpha^*), \tag{9}$$

where $\alpha_{\text{def}}$ is the model's default (unmodified) attention pattern, and $\hat{\alpha} = \alpha_{\hat{\tau}}$ is the best sub-optimal attention pattern found by DTP.

**Interpretation.** This framework provides a principled way to: (i) quantify how different attention patterns influence action certainty, (ii) compare the model's default attention to DTP-refined attention, and (iii) define a theoretical ceiling on performance achievable *without* altering model architecture or retraining.

Under this formulation, Section 4.2 analyzes how DTP moves the model toward this upper bound by reducing uncertainty through improved visual attention allocation. We evaluate the effect of pruning distracting tokens on SIMPLER WidowX Robot tasks across three VLA models. The results in Figure 4 illustrate how DTP alters the visual attention patterns and allows the models to approach their performance upper bounds. For **Nora**, pruning distracting tokens yields substantial improvements across all tasks, despite its relatively weak baseline. **SpatialVLA** exhibits moderate baseline performance and shows relatively consistent gains with pruning. **UniVLA**, while already achieving strong baselines, still benefits from pruning under certain $\tau$ values, highlighting the robustness of our method. Overall, these results confirm that task-specific sub-optimal tolerance values $\hat{\tau}$ exist for each model, with some tasks achieving nearly $2\times$ higher success rates, effectively pushing performance toward the theoretical upper bound under the current architecture.

## 4.3 ABLATION STUDIES

To validate the effectiveness of our DTP framework, we conduct comprehensive ablation studies comparing different token selection strategies within the DTP framework to demonstrate the superiority of our targeted distracting token identification approach. **Random_all_region:** Randomly

Table 4: **Ablation study on WidowX Robot tasks**. We compare different token selection strategies within our DTP framework including random pruning, random pruning only in unimportant region, and no Gaussian methods. Results demonstrate the superiority of our targeted distracting token identification approach.

| Method | Put Spoon on Towel | | Put Carrot on Plate | | Stack Green Block on Yellow Block | | Put Eggplant in Yellow Basket | | SR | Rel. |
|---|---|---|---|---|---|---|---|---|---|---|
| | Grasp | Success | Grasp | Success | Grasp | Success | Grasp | Success | | |
| SpatialVLA + DTP (Random_all_region) | 16.7% | 12.5% | 25.0% | 20.8% | 33.3% | 8.3% | 37.5% | 20.8% | 15.6% | 53.4% |
| SpatialVLA + DTP (Random_unimportant_region) | 12.5% | 8.3% | 20.8% | 4.2% | 41.7% | 25.0% | 37.5% | 33.3% | 17.7% | 60.6% |
| SpatialVLA + DTP (No_Gaussian) | 12.5% | 12.5% | 37.5% | 20.8% | 70.8% | 25.0% | 75.0% | 58.3% | 29.2% | 100.0% |
| SpatialVLA + DTP (Ours) | 25.0% | 25.0% | 45.8% | 33.3% | 54.2% | 29.2% | 62.5% | 62.5% | 37.5% | 128.4% |
| Nora + DTP (Random_all_region) | 37.5 | 8.3% | 41.7% | 0.0% | 37.5% | 4.2% | 0.0% | 0.0% | 3.1% | 50.0% |
| Nora + DTP (Random_unimportant_region) | 37.5% | 12.5% | 50.0% | 4.2% | 58.3% | 4.2% | 4.2% | 0.0% | 5.2% | 83.9% |
| Nora + DTP (No_Gaussian) | 41.7% | 8.3% | 41.7% | 4.2% | 41.7% | 4.2% | 0.0% | 0.0% | 4.2% | 67.7% |
| Nora + DTP (Ours) | 37.5% | 20.8% | 41.7% | 4.2% | 45.8% | 12.5% | 8.3% | 8.3% | 11.5% | 181.7% |
| UniVLA + DTP (Random_all_region) | 50.0% | 41.7% | 58.3% | 54.2% | 91.7% | 33.3% | 95.8% | 83.3% | 53.1% | 77.3% |
| UniVLA + DTP (Random_unimportant_region) | 83.3% | 62.5% | 70.1% | 62.5% | 83.3% | 29.2% | 95.8% | 95.8% | 62.5% | 91.0% |
| UniVLA + DTP (No_Gaussian) | 83.3% | 75.0% | 75.0% | 62.5% | 95.8% | 33.3% | 100.0% | 100.0% | 67.7% | 98.5% |
| UniVLA + DTP (Ours) | 87.5% | 75.0% | 79.2% | 75.0% | 100.0% | 45.8% | 100.0% | 100.0% | 74.0% | 107.7% |

Table 5: **Ablation study on Google Robot tasks**. Comparison of different token selection strategies within our DTP framework across different tasks and variations. Our targeted approach consistently outperforms alternative selection methods across all model variants.

| Method | Visual Matching | | | | Variant Aggregation | | | |
|---|---|---|---|---|---|---|---|---|
| | Pick Coke Can | Move Near | SR | Rel. | Pick Coke Can | Move Near | SR | Rel. |
| SpatialVLA + DTP (Random_all_region) | 62.5% | 57.5% | 60.0% | 75.6% | 66.5% | 52.2% | 59.4% | 74.7% |
| SpatialVLA + DTP (Random_unimportant_region) | 86.0% | 68.8% | 77.4% | 97.5% | 54.2% | 65.8% | 60.0% | 75.5% |
| SpatialVLA + DTP (No_Gaussian) | 83.6% | 75.9% | 79.8% | 101.0% | 89.9% | 68.3% | 79.1% | 99.5% |
| SpatialVLA + DTP (Ours) | 86.7% | 77.1% | 81.9% | 103.1% | 90.2% | 70.7% | 80.5% | 101.3% |
| Nora + DTP (Random_all_region) | 12.1% | 8.9% | 10.5% | 21.6% | 7.4% | 11.2% | 9.3% | 18.5% |
| Nora + DTP (Random_unimportant_region) | 35.2% | 28.7% | 32.0% | 65.6% | 22.1% | 31.4% | 26.8% | 53.1% |
| Nora + DTP (No_Gaussian) | 53.7% | 43.8% | 48.8% | 100% | 56.0% | 44.7% | 50.4% | 100% |
| Nora + DTP (Ours) | 54.7% | 45.0% | 49.9% | 102.4% | 56.7% | 47.0% | 51.9% | 103.0% |

selects and prunes tokens from the entire region of the image. **Random_unimportant_region:** Randomly selects and prunes tokens only within the unimportant region, while preserving tokens from the important region. **No_Gaussian:** Constructs the important region without corner token suppression and Gaussian smoothing. **Ours:** The proposed DTP method with full implementation.

As shown in Table 4 and 5. Across both WidowX and Google Robot tasks, random pruning strategies significantly underperform, often leading to degraded success rates. Restricting randomness to unimportant regions mitigates the drop but remains unstable. No_Gaussian variants achieve better results, yet still fall short of the original model and our full approach, underscoring the benefit of corner suppression and Gaussian smoothing in refining important regions. In contrast, our targeted DTP consistently achieves better outcomes across all models and platforms, validating the importance of precise distracting token identification for robust improvements.

## 4.4 ANALYSIS OF ATTENTION IN UNIMPORTANT REGION AND TASK SUCCESS

To better understand the role of unimportant region attentions in task performance. We define the *unimportant attention* as the sum of attention for all visual tokens in $A_u$ (see Figure 2). This value quantifies the degree to which the model allocates attention to visually irrelevant information during action generation. We conduct experiments on the SIMPLER Benchmark, evaluating both Google Robot and WidowX Robot tasks. For each step, we let the model to generate action tokens with its default visual attention patterns, then we observe and collect unimportant attention values. Finally we group the episodes into *success* and *failure* categories, and we use *Mann-Whitney U test* to examine whether the unimportant attention values differ significantly between them.

As shown in Fig. 5 (top-left), failure episodes consistently exhibit higher unimportant attention than success episodes across all three models ($p < 0.001$). This indicates that attention leakage into irrelevant regions is strongly linked to task failure, and the effect generalizes across architectures. The temporal plots (Fig. 5, right and bottom) unpack the distributions of unimportant attention into a dynamic view across episode steps. While failure episodes generally assign more attention in unimportant region than success episode, the disparity most evident in the middle phase of the trajectory when the robot begins grasping and manipulating the target object. This may suggest a fundamental source of failure in certain robotic manipulation tasks.

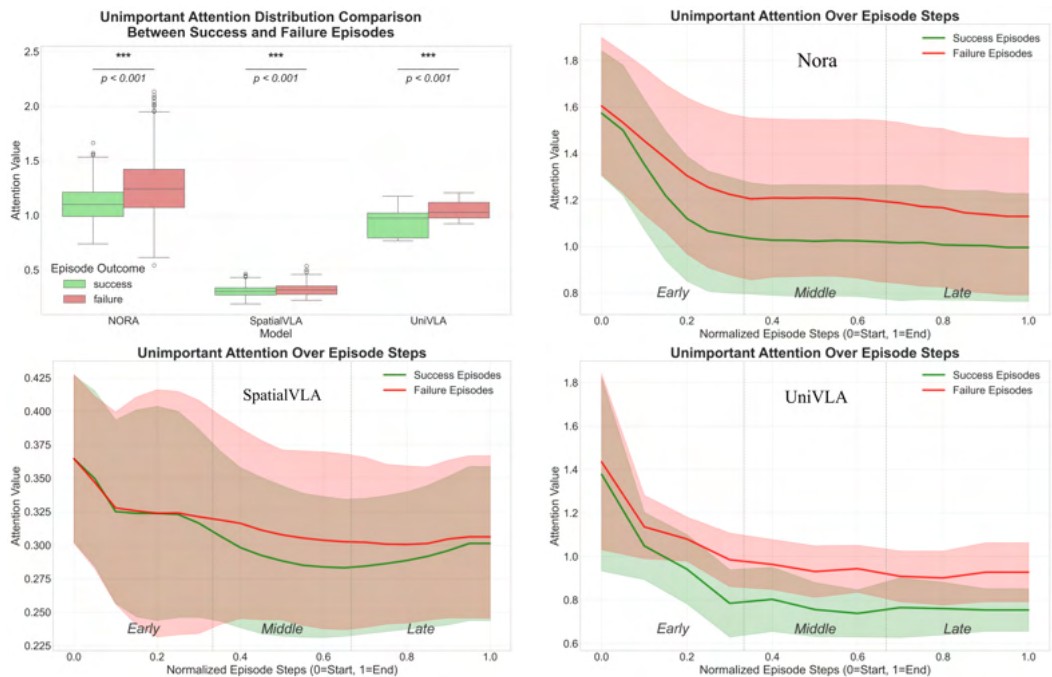

Figure 5: **Unimportant attention in VLA models.** (Top-left) Distribution of unimportant attention values between success and failure episodes across three models. (Top-right, Bottom-left, Bottom-right) Temporal evolution of unimportant attention for each model, with episode steps normalized to $[0, 1]$ to provide a unified view of the dynamic changes.

Overall, these results reveal that the suppression of unimportant attention emerges as a key factor for robust execution, whereas failures tend to persist with high unimportant attention from the middle stage of the trajectory. This finding highlights a systematic weakness of current VLA models: their tendency to misallocate attention to task-irrelevant regions, which not only reduces task success rate but also introduces instability during critical manipulation phases. Importantly, the consistency of this effect across different architectures and robot platforms suggests that it is not model-specific, but rather a general property of vision-language-action pipelines. By quantifying and analyzing unimportant attention, we provide empirical evidence that reducing attention leakage can serve as a practical pathway to improving both efficiency and reliability. These insights also motivate the design of targeted methods, such as our Distracting Token Pruning framework, which explicitly suppresses irrelevant attention and allows the model to focus on task-relevant regions. Ultimately, understanding and controlling unimportant attention offers a principled direction for creating more generalizable VLA models.

## 5 CONCLUSION

In this work, we introduced Distracting Token Pruning (DTP), a simple yet effective plug-and-play framework for evaluating Vision-Language-Action models. By dynamically detecting and pruning distracting tokens in task-irrelevant regions, DTP corrects the visual attention patterns for more accurate action token generation. Our method benefits both strong and weak policies, generalizes across robots. Experiments on the SIMPLER Benchmark and LIBERO Benchmark demonstrated consistent improvements in task success rates. Additionally, with appropriate tolerance $\tau$ settings, we can explore the performance upper bounds of the model under its original architecture. Ablation studies further confirmed the necessity of our targeted pruning strategy over random or simplified baselines. Finally, our analysis revealed a negative correlation between unimportant attention and task success, highlighting attention leakage as a common cause of failure in robotic manipulation. These findings provide new insights into building more robust embodied AI systems.

REPRODUCIBILITY AND OPEN-SOURCE RELEASE

We have open-sourced our code and provided all necessary resources to ensure the reproducibility of our experiments. The released repository includes implementations of our methods, experimental scripts, and figure-generation code, enabling the community to fully reproduce our results and analyses.

ETHICS STATEMENT

This work focuses on advancing the robustness and interpretability of Vision-Language-Action models for robotic manipulation. Our experiments are conducted entirely in simulation, without any involvement of human subjects or sensitive data. We are mindful of the potential societal impacts of embodied AI, particularly in safety-critical applications, and emphasize that our method is intended for research purposes only. Future work that may deploy this method in real-world environments should carefully assess risks such as unintended behaviors or misuse, and implement appropriate safety measures.

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

# A APPENDIX

## ACKNOWLEDGMENTS OF LLM USAGE

We used large language models (LLMs) solely to assist with polishing the writing and improving readability. All research ideas, experimental results, figures, and the substantive content of this paper are entirely our own responsibility.

## A.1 REMARKS FOR SECTION 4.2

In practice, we assume that multiple ground-truth actions $A^*$ may lead to task success, which ensures $H(A^*) > 0$ and makes our normalized formulation well defined. The domain of $\alpha$ corresponds to the cross-attention weights from the output (action) token query to the visual token keys, i.e., the distribution of attention over visual tokens. Here $Z$ denotes the value vectors of visual tokens, and $Z_\alpha$ represents the attention-weighted aggregation used in generating the output token. Different tolerance values $\tau$ prune varying numbers of tokens in the unimportant region (smaller $\tau$ prunes more aggressively), thereby modifying the attention weights $\alpha$ and inducing new attended representations $Z_{\alpha_\tau}$. We seek the tolerance setting that minimizes the conditional uncertainty $H(A^* \mid Z_{\alpha_\tau})$, which corresponds to the model's most effective (sub-optimal) attention pattern under pruning. Since VLA models generate discrete action tokens autoregressively, the conditional entropy $H(A^* \mid Z_\alpha)$ is naturally computed from the predictive distribution over actions. Finally, while the theoretical optimum $\alpha^*$ may be unattainable in practice, our method ensures that the selected $\hat{\alpha} = \alpha_{\hat{\tau}}$ approximates the best possible sub-optimal attention pattern, aligning closely with the model's own preferences and yielding the maximum possible improvements in task success rate.

## A.2 IMPLEMENTATION DETAILS

Our method is implemented as a plug-and-play module that integrates seamlessly with existing VLA architectures. We register forward hooks on attention layers to extract attention scores without modifying the model architecture. The method operates during inference with negligible overhead, requiring only one additional forward pass for generating a single token to compute visual proportion weights.

For experiments in Section 4.1, we used a fixed tolerance score of $\tau = 0.5$ (for SpatialVLA (Qu et al., 2025)), $\tau = 1.22$ (for Nora (Hung et al., 2025)), and $\tau = 0.7$ (for UniVLA (Wang et al., 2025)) for all tasks.

For SpatialVLA with 256 visual tokens, the selected layers for constructing the relevance heatmap are empirically chosen as $\mathcal{C} = 4, 6$ (indexed from layer 0) based on their strong visual-linguistic alignment properties. We set $k = 109$ for selecting the top-relevant image tokens from relevance heatmap, and we set $\sigma = 0.65$ for Gaussian smoothing when constructing the final important region.

For Nora with 64 visual tokens, the selected layers for constructing the relevance heatmap are empirically chosen as $\mathcal{C} = 12, 13, 21$ (indexed from layer 0) based on their strong visual-linguistic alignment properties. We set $k = 40$ for selecting the top-relevant image tokens from relevance heatmap, and we set $\sigma = 0.65$ for Gaussian smoothing when constructing the final important region.

For UniVLA with 1024 visual tokens, the selected layers for constructing the relevance heatmap are empirically chosen as $\mathcal{C} = 11, 12$ (indexed from layer 0) based on their strong visual-linguistic alignment properties. We set $k = 512$ for selecting the top-relevant image tokens from relevance heatmap, and we set $\sigma = 0.9$ for Gaussian smoothing when constructing the final important region.

In our experiments, we use DPT to detect distracting tokens during the model's original action generation, then construct a pruning mask over the visual input, and finally let the model re-generate a refined action token under the masked attention. For SpatialVLA, however, we simplify this process by analyzing only the *first* action token to construct a single pruning mask, which is then applied across all subsequent tokens at the step. This strategy significantly mitigate the inference speed issue, with only a minor trade-off in the precision of distracting token pruning. For Nora, we limit pruning to at most *two* visual tokens, since it operates with relatively fewer visual tokens, and excessive pruning may introduce errors in the generation process. For SIMPLER Benchmark, we sample test cases for Pick Coke Can Variant Aggregation test suite, as it is designed to evaluate efficiency while maintaining overall testing quality and validity of results.

In the experiments of Section 4.2, we use an interval of $\tau = 0.1$ for empirical evaluation on the SIMPLER WidowX Robot tasks to identify the optimal $\hat{\tau}$. For UniVLA, however, we adopt a coarser interval of $\tau = 1.0$ once $\tau > 3.0$, since smaller increments beyond this point yield negligible differences in task success rate, and the larger step size improves experimental efficiency.

All experiments for SpatialVLA and Nora were conducted on a single NVIDIA RTX 3090 GPU, while UniVLA was run on an NVIDIA A100 GPU. (For reproducibility, we recommend using the same GPU setup, as different GPU may lead to slight variations in the results.)

### A.3 LIMITATION

While DTP demonstrates strong generalizability to transformer-based VLA models and serves as an effective approach to explore performance upper bounds without altering model architectures, our implementation has some limitations. First, the current system could potentially be optimized to achieve faster inference speed, which would improve its efficiency. Second, the construction of important areas relies solely on the model's own ability, with few rule-based refinements. This dependence may limit precision, and future work could incorporate alternative methods for generating more accurate important regions, thereby further enhancing the overall performance of the framework.

## A.4 ADDITIONAL VISUALIZATION CASES OF OUR METHOD

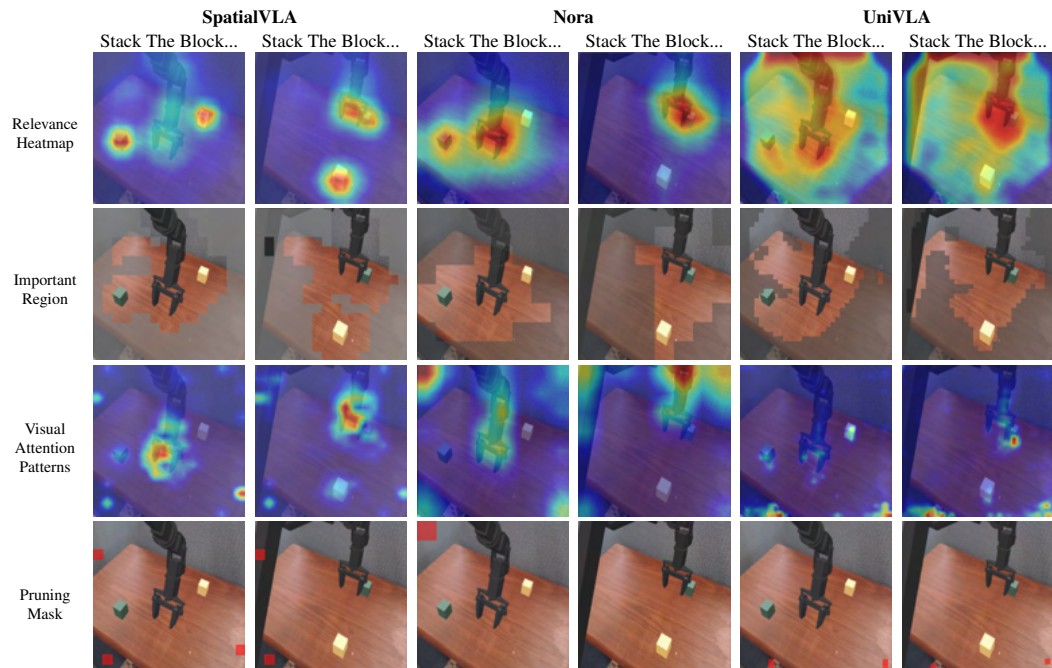

Figure 6: **Visualization of the distracting token pruning process across different VLA models for the stacking task.** The figure shows the Relevance Heatmap (first row), Important Region (second row), Visual Attention Patterns (third row), and Final Pruning Mask (last row) for three VLA models (SpatialVLA (Qu et al., 2025), Nora (Hung et al., 2025), UniVLA (Wang et al., 2025)). The comparison reveals how our method identifies and prunes distracting visual tokens across different model architectures.

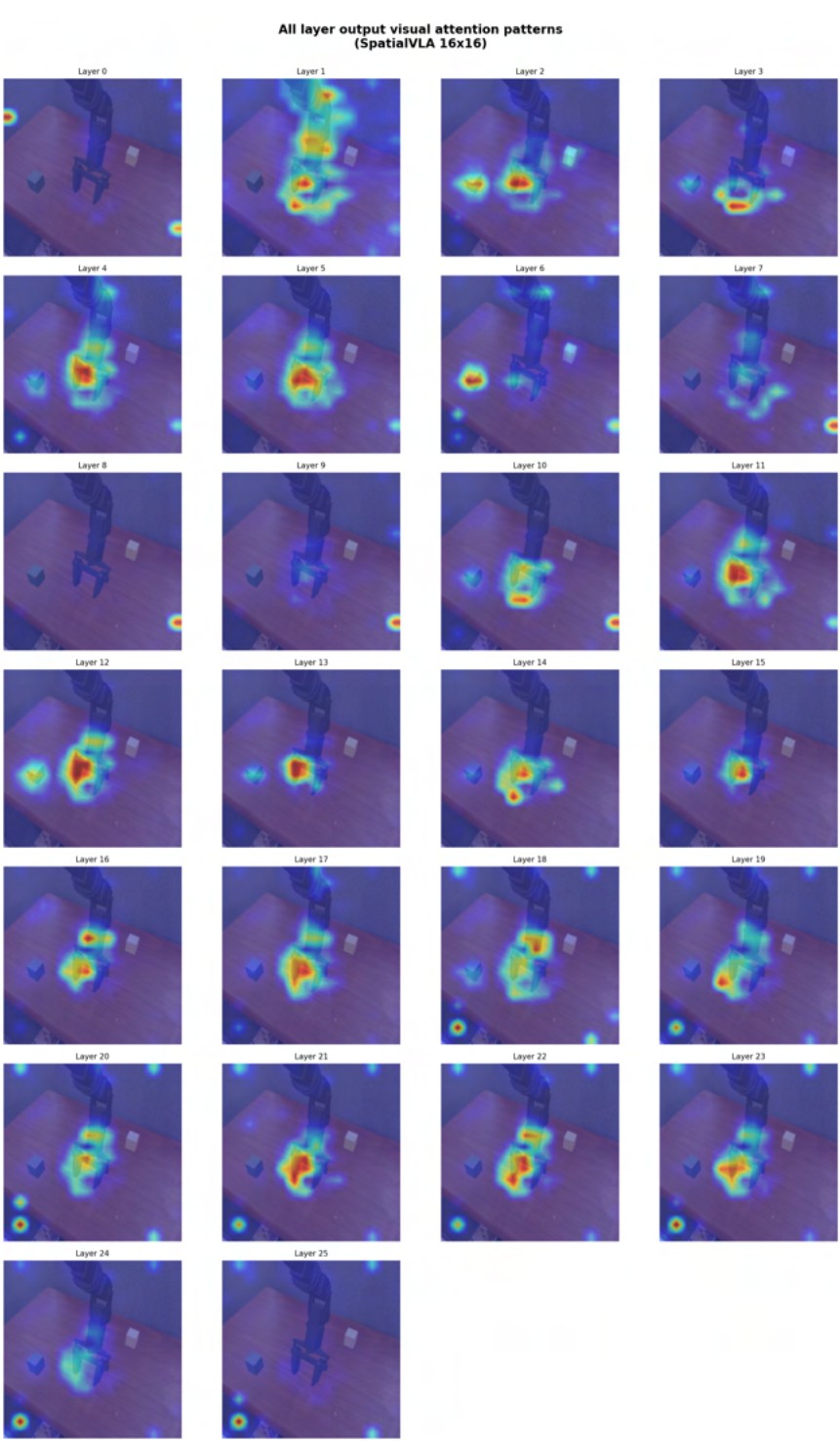

Figure 7: **Layer-wise visual attention patterns of SpatialVLA for the task 'Stack the Block...'**
Each image shows the attention distribution at a different transformer layer when the model generates the action token.

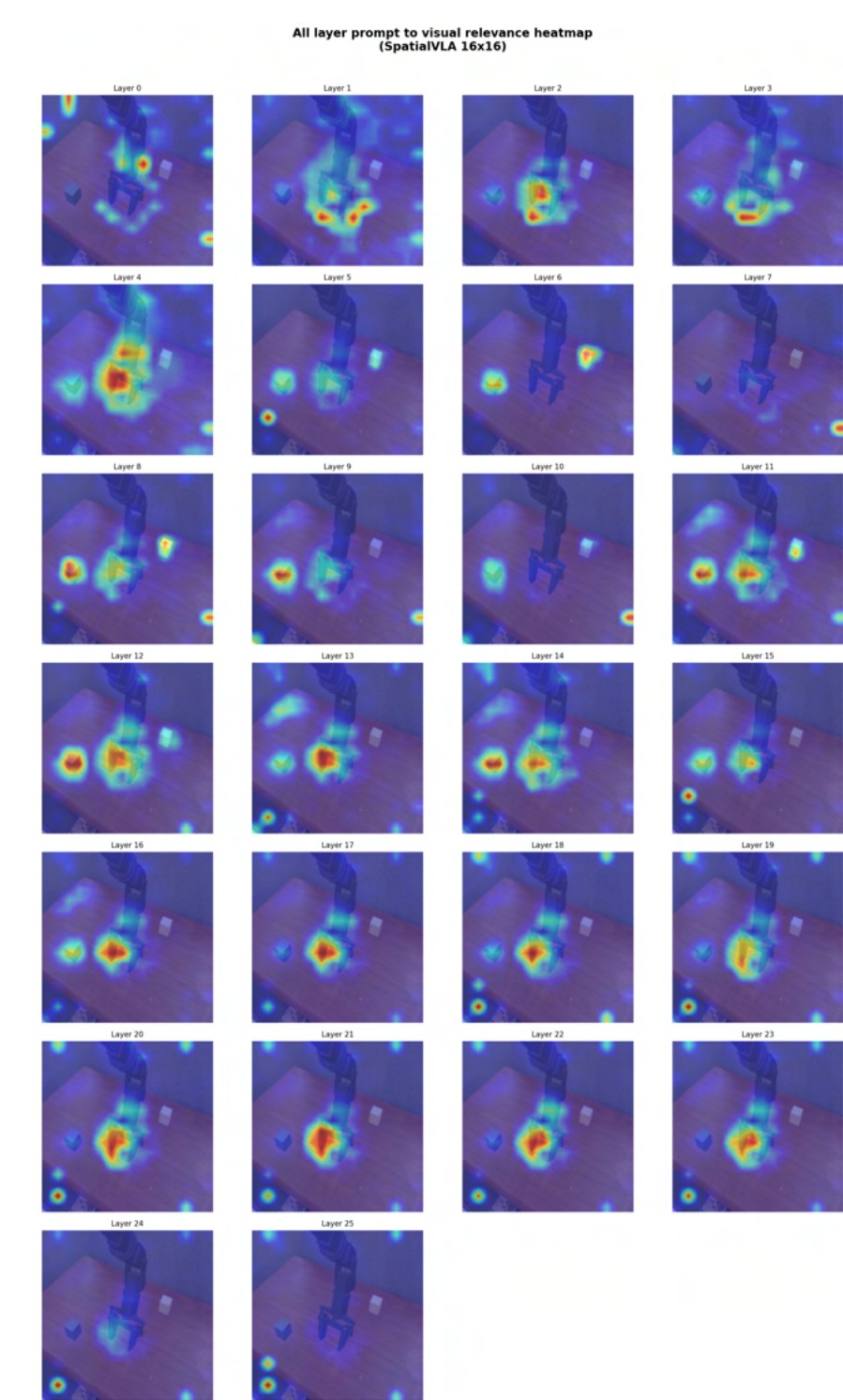

Figure 8: **Layer-wise relevance heatmap of SpatialVLA for the task 'Stack the Block...', with selected layer 4, 6 used for constructing important region.**

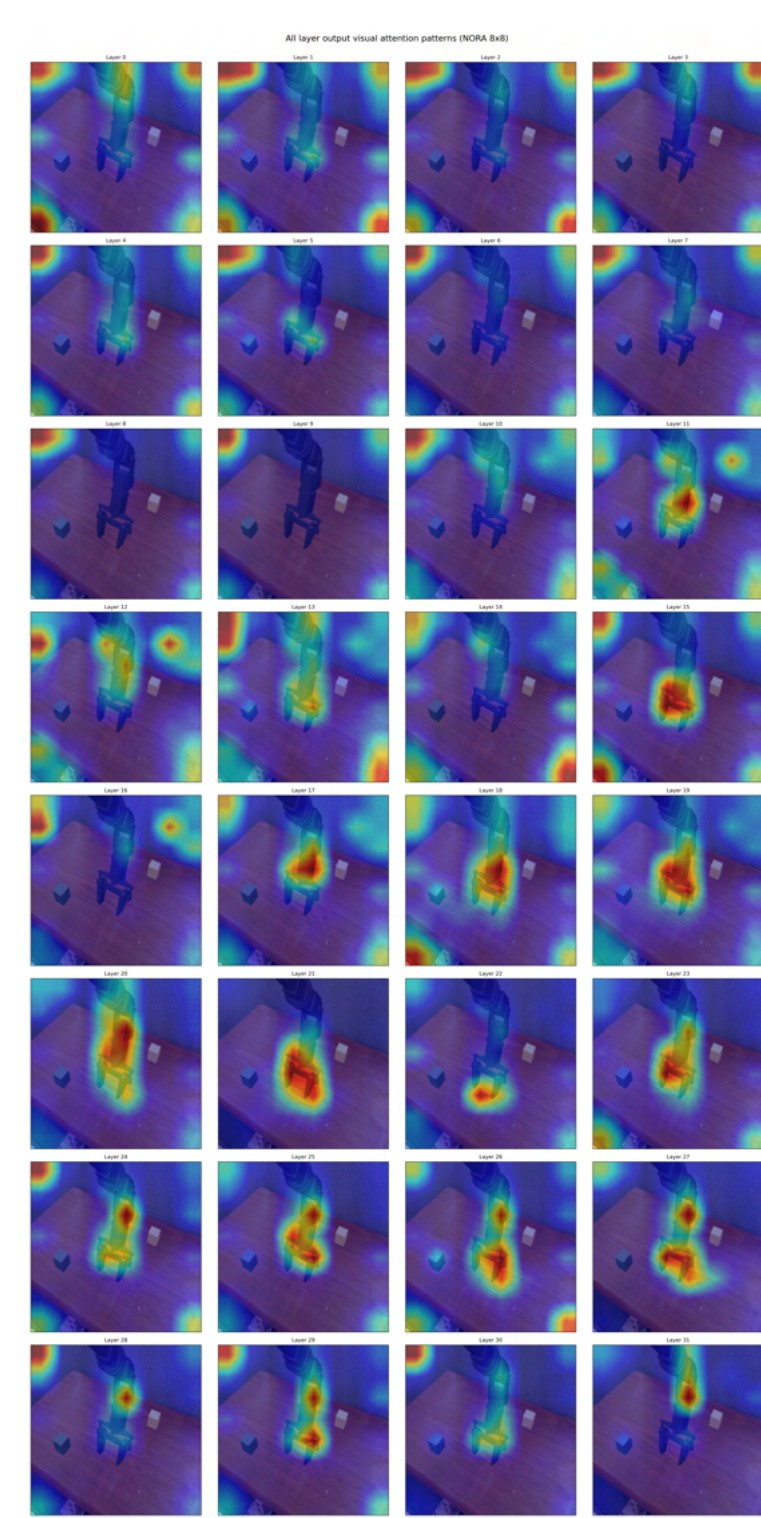

Figure 9: **Layer-wise visual attention patterns of Nora for the task 'Stack the Block...'** Each image shows the attention distribution at a different transformer layer when the model generates the action token.

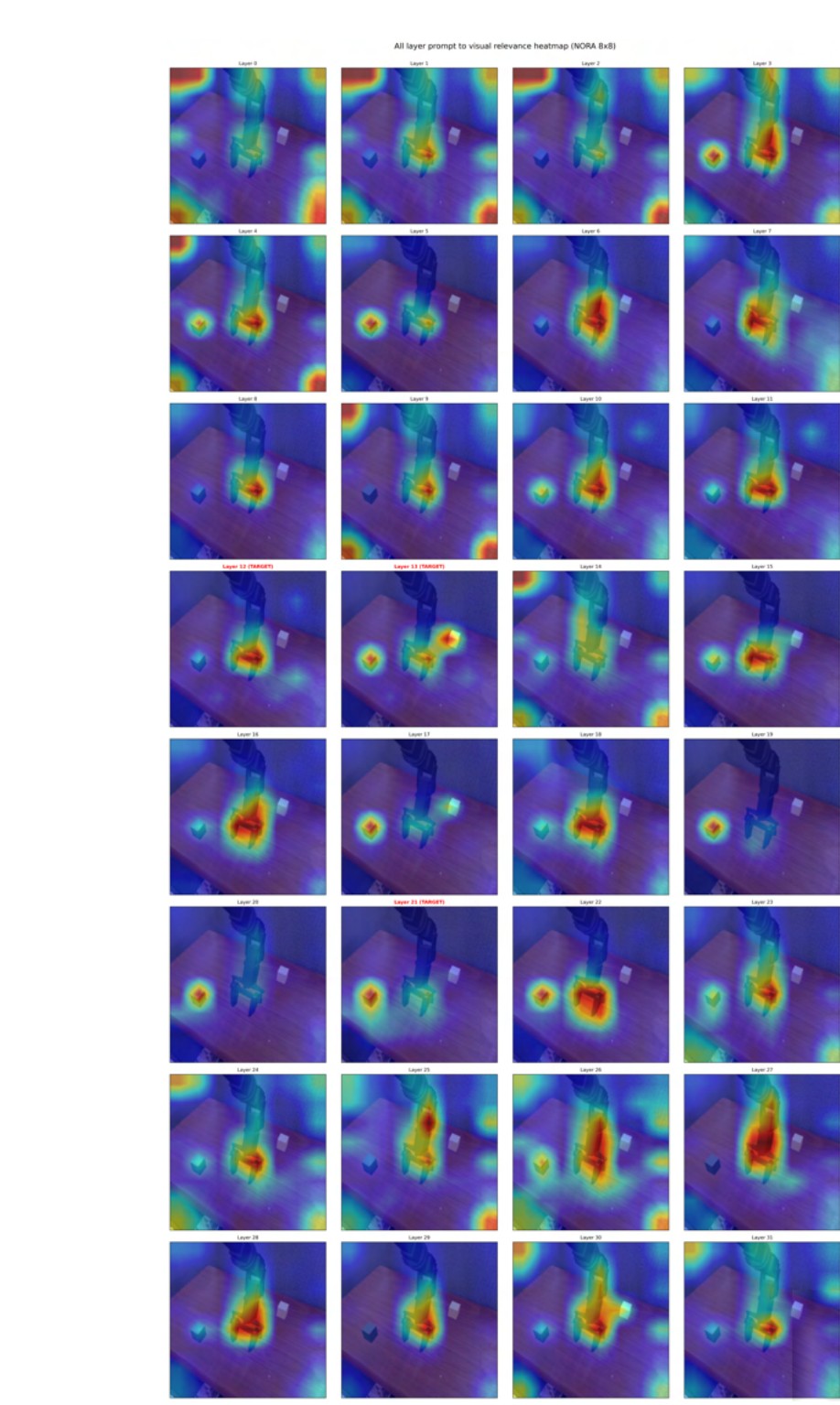

Figure 10: **Layer-wise relevance heatmap of Nora for the task 'Stack the Block...', with selected layer 12, 13, 21 used for constructing important region.**

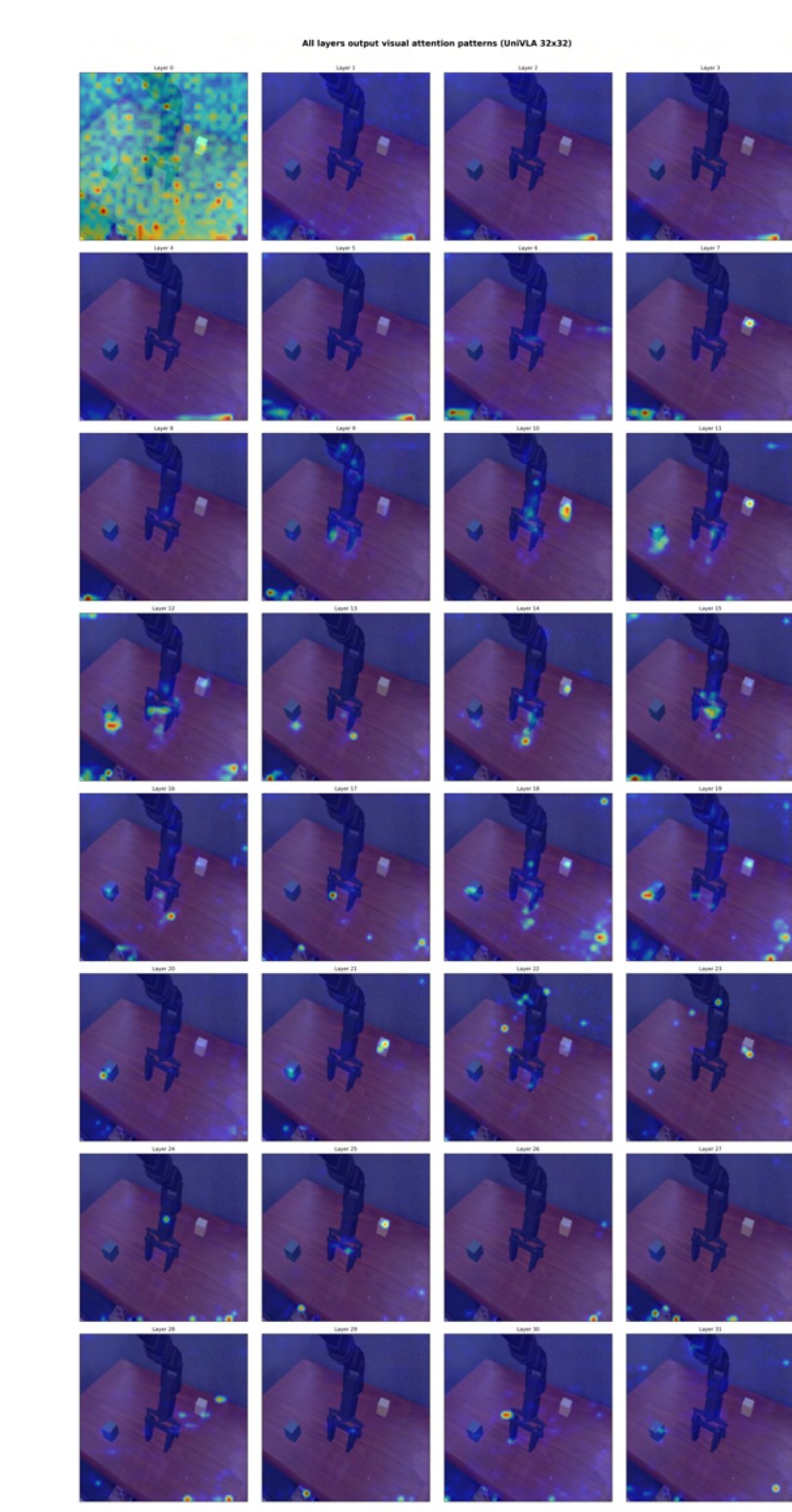

Figure 11: **Layer-wise visual attention patterns of UniVLA for the task 'Stack the Block...'** Each image shows the attention distribution at a different transformer layer when the model generates the action token.

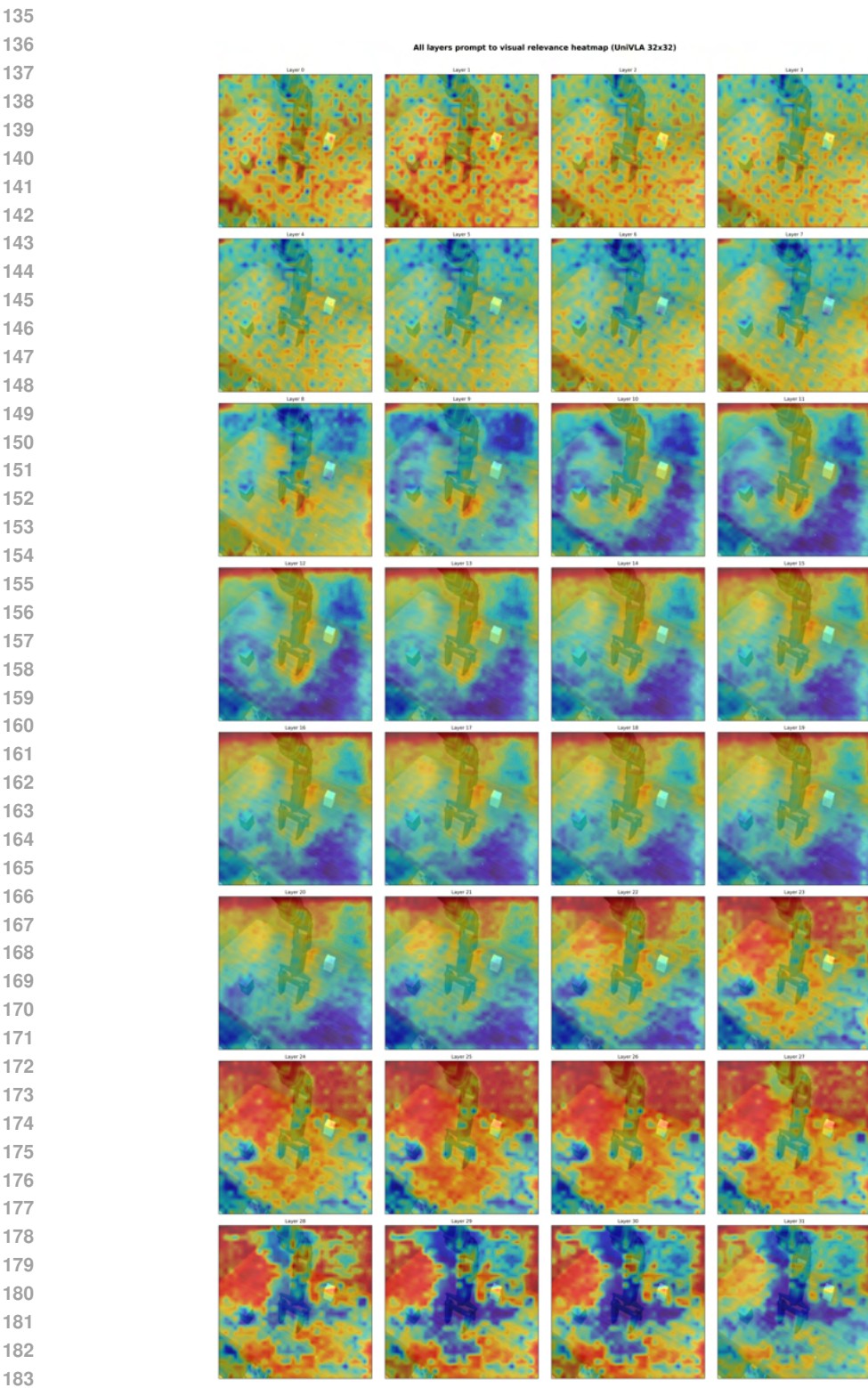

Figure 12: **Layer-wise relevance heatmap of UniVLA for the task 'Stack the Block...', with selected layer 11, 12 used for constructing important region.**

