# OpenReview forum: "DTP: A Simple yet Effective Distracting Token Pruning Framework for Vision-Language Action Models"
_ICLR.cc/2026/Conference — Submitted to ICLR 2026_

### Official Review · Reviewer_3oSv · 2025-10-20

**Soundness:** 2
**Presentation:** 3
**Contribution:** 2
**Rating:** 4
**Confidence:** 4

**Summary:**

Existing VLAs may overly attend to task-irrelevant vision tokens during inference, ultimately hurting policy performance. The authors propose distracting token pruning (DTP), a framework to dynamically detect and prune the distracting tokens. Experiments are conducted purely in simulation, in SIMPLER, and show minor improvements over baselines (SpatialVLA, Nora, UniVLA) without DTP.

**Strengths:**

The proposed DTP framework is techincally sound and the simulations appear well executed. The authors test DTP across three different VLAs, demonstrating their approach is model agnostic.

A particular strength lies in the ablations (Tables 3 and 4). The improvement over the random pruning ablation is insightful, which shows DTP's efficacy.

Finally, the paper is very clear. The problem is well motivated and the proposed solution is explained well. The attention visuals are clean and instructive.

**Weaknesses:**

Exclusive reliance on simulation: there are no real-world robotic experiments. While simulators like SIMPLER may serve as a proxy for downstream deployment, the results obtained therein are not always indicative of the real-world, and any conclusions we draw from simulation must be tempered. The sim-to-real-gap is notorious challenging, and by confining all analysis to simulation, the paper's core claims lack contextualization on real hardware, which is ultimately where we want to see improvement benefits.

Marginal performance gains on baselines: the simulation results, while consistently positive, do not result in large improvements. For example, averaging the results (as exemplified by Figure 1 and the Tables), the relative improvement with DTP is not very great. If these results were contrasted against real-world experiments, I think modest improvements are justified, but in simulation, the bar is higher (because the results don't always translate across the sim-to-real gap). This raises questions about the method's utility.

**Questions:**

Comparison to prior work: while this work proposes mechanistic interpretability techniques applied to VLAs, the core idea of a run-time intervention scheme is not entirely new, and contextualizing this work with regard to prior work would be beneficial. For instance, [1] proposes a run-time intervention scheme to alter task-irrelevant regions in the VLA's observation based upon sensitivity. For offline work, [2] presents an architecture to filter visual features to focus on task-relevant information. It would be great to add [2] as a possible baseline for comparison. It makes more sense to discuss these types of works, instead of VLMs, in related work.

Threshold parameter $\tau$. Figure 4 shows that $\tau$ is task and model dependent, in addition to be sensitive to value. This seems to challenge the generality of the proposed approach. How do you envision a roboticist using DTP in the real-world? Would they select this hyperparameter for every new task, or do you think a single value would suffice and be task-invariant?

Inference time overhead: the appendix (pg. 13, line 697) states that tricks were applied to SpatialVLA to mitigate inference speed issue. Could you elaborate on this point? It seems otherwise the inference speed may be non-negligible.

Generality of Important Region:  DTP applies corner suppression and Gaussian smoothing to construct the import regions. While an ablation is provided, this design choice seems to implicitly limit DTP to simple tasks. Do you think this part of the method would work if the key object were located in the corner of the observation, for example?

Number of evaluations: for the main results in Tables 1 and 2, can you please clarify the number of rollouts performed for each task. I couldn't find this information in the main part of the manuscript and is important for assessing the statistical significance of the reported improvements.

[1] Hancock, A. J., Ren, A. Z., & Majumdar, A. (2025, May). Run-time observation interventions make vision-language-action models more visually robust. In 2025 IEEE International Conference on Robotics and Automation (ICRA) (pp. 9499-9506). IEEE.
[2] Huang, H., Liu, F., Fu, L., Wu, T., Mukadam, M., Malik, J., ... & Abbeel, P. (2025). Otter: A vision-language-action model with text-aware visual feature extraction. arXiv preprint arXiv:2503.03734.

---

> ### Author Response · Authors · 2025-12-03
>
> ###  **Response to Weaknesses 1: Results only on SimplerEnv.**
>
> Thank you for raising this important point. We fully agree that real-world experiments are importnat standard for evaluating robotic policies, and that the sim-to-real gap is a long-standing challenge. Our decision to focus on simulation is deliberate and aligned with the scope and purpose of this work. Below, we clarify why our evaluation setting is still meaningful and appropriate.
>
> **1. Our contribution targets a model-level phenomenon, confirmed across two diverse simulation benchmarks**
>
> The core objective of this work is to **reveal and correct a model-internal failure mode in current VLAs: their visual attention often concentrates on task-irrelevant tokens, even when the model is otherwise strong. This is a fundamental issue arising from internal attention dynamics, not from a particular simulation environment.**
>
> Importantly, we validate this phenomenon, and DTP’s corrective effect across two structurally different simulation benchmarks:
>
> * SimplerEnv: validated to correlate strongly with real-world robot performance, with realistic visual conditions, distribution shifts, and environment setups.
>
> * LIBERO: a more diverse benchmark with complex, multi-stage manipulation tasks and different environment structures.
> | Method | LIBERO-Spatial | LIBERO-Object | LIBERO-Goal | LIBERO-10 |
> |------------|----------------------|-----------------------|--------------------|-----------------|
> | Nora| 92.8% |76.0%| 85.8%|69.6%|
> | **Nora + DTP (Ours)**| **93.0%** | **77.4%** |**88.4%** |**76.2%** |
> |
>
> Across both benchmarks, and across all three VLAs we evaluate, we consistently observe:
>
> 1. the same attention misalignment failure mode, and
>
> 2. consistent improvements after applying DTP.
>
> **This cross-benchmark consistency strongly supports that the issue and the solution, are model-level, not simulator-specific. The fact that DTP generalizes across different architectures, task distributions, and environment variations further indicates that the mechanism we propose is not tied to any specific simulated setting, but instead addresses a deeper structural weakness of current VLA systems.**
>
> **2. SIMPLER is explicitly designed to reflect real-world robot performance, making it a meaningful and reliable evaluation setting**
>
> We fully understand the reviewer’s concern about relying on simulation, and we emphasize that our choice of SIMPLER is not arbitrary. **SIMPLER was specifically engineered to address the sim-to-real gap, and the benchmark’s creators provide extensive evidence that policy behavior in SIMPLER strongly correlates with real-world execution.**
>
> More concretely, the SIMPLER paper [1] stated that:
>
> * **Simulation-based evaluation can be a scalable, reproducible, and reliable proxy for real-world evaluation**
>
> * **There is strong correlation between policy performance in SIMPLER environments and in the real world.**
>
> * **SIMPLER evaluations accurately reflect real-world policy behavior modes such as sensitivity to various distribution shifts.**
>
> Instead of building expensive, full-fidelity digital twins which quickly become infeasible as tasks diversify, SIMPLER introduces targeted techniques to mitigate exactly these disparities, including:
>
> * realistic visual domains modeled after real robot setups,
>
> * simulated camera models matching those in real hardware,
>
> * carefully calibrated control parameters aligned with real robot behavior,
>
> * reconstruction of environments used in real-world benchmarks (e.g., table geometry, object placements), and
>
> * procedures for constructing new environments that preserve sim-to-real consistency.
>
> Thus, SIMPLER is not merely a generic simulator, it is one of the most rigorously validated simulation frameworks available for studying real-world robot policies, especially in the context of perception-driven manipulation, where visual fidelity and attention distribution are critical.
>
> Because our work focuses on model-internal visual attention misalignment, the strong sim-to-real fidelity of SIMPLER gives confidence that:
>
> * the misaligned-attention phenomenon we observe is not simulator-specific, and
>
> * the corrective effects of DTP are likely relevant to real-world policy behavior.
>
> Combined with our additional evaluation on the LIBERO benchmark, which has different dynamics, tasks, and complexity—this provides strong evidence that the insights from our study arise from fundamental model limitations, not artifacts of a particular simulated environment.
>
> [1] Xuanlin Li, Kyle Hsu, Jiayuan Gu, Karl Pertsch, Oier Mees, Homer Rich Walke, Chuyuan Fu, Ishikaa Lunawat, Isabel Sieh, Sean Kirmani, Sergey Levine, Jiajun Wu, Chelsea Finn, Hao Su, Quan Vuong, and Ted Xiao. Evaluating real-world robot manipulation policies in simulation, 2024. URL https://arxiv.org/abs/2405.05941.

---

> ### Author Response · Authors · 2025-12-03
>
> ###  **Response to Weaknesses 2: Marginal performance gains on baselines.**
>
> Thank you for this thoughtful observation. We appreciate the concern that improvements in simulation must be interpreted carefully. Below, we clarify why the gains shown in our work are both meaningful and scientifically important.
>
> **1. The baseline VLAs are already near-saturated on many tasks, making even small absolute gains meaningful**
>
> Many tasks in SIMPLER and LIBERO are already solved at 70–95% success by strong VLAs such as UniVLA and Nora.
> In these high-performance regimes:
>
> * **small absolute gains correspond to large relative error reductions, and**
>
> * **incremental improvements become increasingly difficult to achieve, especially without retraining or architecture changes.**
>
> For example:
>
> * On LIBERO-10 (a challenging multi-stage suite), we obtain +6.6% absolute, which corresponds to almost 10% relative improvement in sucess rate.
>
> * On SIMPLER tasks, gains range from +20% to nearly +100% relative increases for weaker models.
>
> Thus, “modest” absolute values under high baseline performance still reflect significant improvements in remaining failure cases, which are often hard for VLAs.
>
> **2. DTP operates under extremely constrained conditions: no retraining, no architecture change, no extra inputs**
>
> Unlike most methods that improve robot manipulation success rates, DTP:
>
> * does not modify the VLA architecture,
>
> * does not use additional data or inputs,
>
> * does not rely on fine-tuning or policy optimization, and
>
> * is applied in a single forward pass.
>
> Under these constraints, achieving any consistent gain across all models and benchmarks is valuable.
> DTP improves performance exclusively by correcting internal visual attention allocation—something current VLAs do not handle well.
>
> Given this strict constraint, the observed improvements demonstrate that:
>
> **very strong VLA models can still suffer from significant visual attention misalignment, and correcting this misalignment alone meaningfully boosts performance.**
>
> This insight is scientifically important independent of the magnitude of task-level improvement.
>
> **3. The improvements are consistent across all models, all tasks, and two distinct benchmarks**
>
> A key strength of DTP is consistency:
>
> * 3 out of 3 VLAs improve (SpatialVLA, Nora, UniVLA).
>
> * Every SIMPLER task improves.
>
> * All four LIBERO suites improve.
>
> * Gains appear in both zero-shot and trained regimes.
>
> Consistency across 6+ environments, dozens of rollouts, and two structurally different benchmarks indicates that DTP addresses a systematic, model-level failure, not noise or overfitting.
>
> Consistency across architectures is arguably more important than large absolute numbers in a single setting.
>
> **4. The value of our method is diagnostic and foundational: it reveals why VLAs fail**
>
> This paper’s primary contribution is not about beating state-of-the-art numbers by a large margin.
> Instead, it uncovers a fundamental issue:
>
> VLA models frequently misallocate attention to task-irrelevant visual tokens, and this misalignment is a major contributor to failure cases. **A notable benefit of DTP is that it provides an interpretability and diagnostic lens: by making the model’s visual attention patterns explicit and then correcting them, DTP helps reveal the underlying causes of VLA errors and how they can be resolved. This supplies a useful tool for analyzing how semantic information flows within VLAs and supports future debugging and research on attention behaviors in embodied systems.**
>
> **In summary, DTP is an insightful and novel method to:**
>
> * **identify this widespread phenomenon empirically,**
>
> * **quantify its impact on performance, and**
>
> * **correct it in a general, training-free, plug-and-play manner.**
>
> * **providing an interpretability and diagnostic lens of why VLA model fail.**
>
> The gains we report are the direct result of fixing only the attention misalignment—without any external help.
> This highlights the untapped potential inside existing VLA architectures.
>
> This diagnostic insight is meaningful even if improvements are not large in absolute terms.

---

> ### Author Response · Authors · 2025-12-03
>
> ###  **Response to Question 1: Comparison to prior work.**
>
> Thank you for pointing out these relevant works and for highlighting the importance of contextualizing our method within the growing literature on run-time interventions and feature-selective architectures. We appreciate the reviewer bringing both [1] and [2] to our attention and agree that they should be discussed in our Related Work section.
>
> Below, we clarify how our work relates to these approaches and why they differ in scope, assumptions, and goals.
>
>
> **1. Relation to [1]: Run-time observation interventions**
>
> Hancock et al. (ICRA 2025) [1] indeed propose a run-time intervention mechanism that modifies observations based on visual sensitivity. Their work focuses on input-space manipulation (e.g., masking or altering pixel regions) to improve robustness under distribution shifts.
>
> Our method differs in two key ways:
>
> * **Internal vs. external intervention**: DTP operates directly on the model's internal attention representations, rather than modifying the raw observations. We analyze and correct the VLA’s visual-token attention allocation, which **reveals a model-level failure mode rather than an input-level one.**
>
> * **Architectural and training independence**: **DTP requires no retraining, no fine-tuning, and no architectural modification, whereas the effects of [1] depend on the model's sensitivity to input perturbations.**
>
> However, we agree that [1] is relevant and we will integrate a comparison into the related work.
>
> **2. Relation to [2]: Otter—text-aware visual feature extraction**
>
> Huang et al. (2025) [2] propose text-conditioned visual feature filtering as part of the Otter architecture. It differs fundamentally from DTP:
>
> * **Requires model redesign and additional training:**
> Otter introduces a new visual encoder, new feature selection layers, and is trained end-to-end to learn these filters.
> This makes it an architectural method, not a general plug-in applicable to existing VLAs.
>
> * **Not inference-time model-agnostic:**
> Because Otter requires model-specific components and training, it cannot be directly applied to SpatialVLA, Nora, UniVLA models, etc.
> In contrast, DTP is architecture-agnostic, requires no training, and is immediately applicable to any VLA with image tokens.
>
> * **Different objective:**
> Otter aims to improve visual representation learning through architectural improvements.
> Our work aims to reveal a systemic misalignment in existing models’ internal attention: a diagnostic insight that is not addressed by [2].
>
> Thus, while [2] is relevant and will be cited, it is not a suitable baseline for comparison, as it belongs to a separate category of training-based architectural methods, whereas our method is simple and novel, training-free inference-time correction mechanism.
>
> In summary, given these differences, [1] and [2] are not an apples-to-apples baseline for DTP, and comparing them numerically would be misleading. However, we agree that both works enrich the conceptual landscape and we will incorporate them into the Related Work section with clear distinctions.
>
> [1] Hancock, A. J., Ren, A. Z., & Majumdar, A. (2025, May). Run-time observation interventions make vision-language-action models more visually robust. In 2025 IEEE International Conference on Robotics and Automation (ICRA) (pp. 9499-9506). IEEE. [2] Huang, H., Liu, F., Fu, L., Wu, T., Mukadam, M., Malik, J., ... & Abbeel, P. (2025). Otter: A vision-language-action model with text-aware visual feature extraction. arXiv preprint arXiv:2503.03734.

---

> ### Author Response · Authors · 2025-12-03
>
> ###  **Response to Question 2: Clarification on the threshold parameter $\tau$ and its use in real-world settings.**
>
> Thank you for raising this important point. We understand the concern that Figure 4 may misleadingly suggest $\tau$ is task-dependent or sensitive, potentially affecting practical deployment. Below we clarify the intention of Figure 4, the actual usage of $\tau$ in Section 4.1, and how a roboticist would apply DTP in real-world settings.
>
> **1. Clarification: In all main experiments (Section 4.1), we use a single $\tau$ per model across all tasks and environments**
>
> This is a key point that is easy to miss if one only focuses on Figure 4:
>
> * In Section 4.1 (the main evaluation),
> we select one global $\tau$ per VLA (SpatialVLA, Nora, UniVLA).
>
> * This global $\tau$ is used unchanged across all tasks, all rollouts, and all environments (SimplerEnv WidowX, Google Robot, and LIBERO).
>
> * We do not select or tune $\tau$ separately for each task.
>
> Despite this intentionally strict setting, DTP still produces consistent performance improvements across all tested tasks and both benchmarks.
>
> This demonstrates that DTP does not require per-task tuning and that $\tau$ functions as a model-level hyperparameter, not a task-level one.
>
> **2. Why Figure 4 shows task-dependent variation: Section 4.2 serves a different, exploratory purpose**
>
> * Figure 4 is not used to set $\tau$, but to study the model’s potential under different attention patterns.
>
> * Section 4.2 intentionally sweeps $\tau$ over a range to understand
> how the model’s performance changes as visual attention is systematically optimised.
>
> This analysis reveals two important insights:
>
> * Current VLAs often operate far below their attention-based performance potential.
>
> * There exists a sub-optimal attention pattern $\hat{\alpha}$ that can push performance toward the model’s upper bound under its current architecture.
>
> Thus, Figure 4 is diagnostic, not prescriptive.
> It is meant to expose the phenomenon, not to imply that per-task tuning is required in practice.
>
> **3. How would a roboticist use DTP in the real world?**
>
> A practitioner would not tune $\tau$ per task.
> The real-world workflow is simple:
>
> * Use a small validation set (e.g., a few rollouts or a small dataset of representative scenes).
>
> * Select one global $\tau$ for the VLA model (e.g., for UniVLA or SpatialVLA).
>
> * Deploy the model with that fixed $\tau$ across all downstream tasks.
>
> **Because $\tau$ is model-level and not task-level, the cost of adopting DTP in practice is very low.**
>
> **Our experiments show that once $\tau$ is selected once per model, it generalizes robustly across:**
>
> * SimplerEnv task suite (different object placements, lighting, backgrounds, etc)
>
> * LIBERO benchmark (different tasks, dynamics, long-horizon sequences)
>
> * Zero-shot & trained models (SpatialVLA, Nora, UniVLA)
>
> This gives strong empirical evidence that a single $\tau$ is sufficient and stable across environments.
>
> **4. Why $\tau$ is robust in practice despite variation shown in Section 4.2**
>
> Although Figure 4 shows different tasks peaking at different values, this does not contradict robustness:
>
> * Section 4.2 uses fine-grained sweeping to study how far performance could be pushed.
>
> * Even though sub-optimal peaks differ, the performance plateau around global $\tau$ remains wide and stable.
>
> * The “optimal” $\hat{\tau}$ simply shows the theoretical performance potential, not the required setting.
>
> This is similar to choosing a single learning rate for an entire model, even though different layers might theoretically prefer different values.
>
> **To summarize:**
>
> * **In practice:
> A roboticist will set one $\tau$ per model, using a few validation rollouts, and that single value will generalize well.**
>
> * **No per-task tuning is needed:
> All main experiments use a single global $\tau$, and the method still shows consistent improvements across all tasks and benchmarks.**
>
> * **Figure 4 is exploratory:
> It reveals the latent performance ceiling of VLAs when internal attention is optimized, not a requirement for tuning.**
>
> * **Real-world applicability:
> Because DTP is training-free, architecture-free, and uses a model-level $\tau$, it is straightforward to deploy on real robots with minimal overhead.**

---

> ### Author Response · Authors · 2025-12-03
>
> ###  **Response to Question 3: Inference time overhead.**
>
> Thank you for raising this point. We appreciate the opportunity to clarify our efficiency design choices and the inference-time overhead introduced by DTP. Our goal from the beginning was to ensure that DTP remains lightweight, practical, and deployable, even though its primary purpose is accuracy improvement, not FLOP reduction.
>
>
> **1. Efficiency trick used in SpatialVLA: compute attention only once, not per action token**
>
> For SpatialVLA, computing the full visual attention heatmap for every generated action token would indeed introduce non-negligible overhead. To avoid this, we implemented the following optimization:
>
> * We compute the visual attention pattern only for the first generated action token.
>
> * We apply distracting token pruning based on this first-step attention pattern.
>
> * The pruned visual attention mask are then reused for all subsequent action tokens in the same rollout.
>
> * No additional attention-tracing is performed afterward.
>
> This reduces the cost of DTP from "per action token" to essentially "per step," which dramatically lowers overhead while still maintaining the success rate improvements reported in Section 4.1.
>
> This is the "efficiency trick" mentioned in the appendix, and it is crucial for making DTP fast enough for real-world deployment.
>
> **2. Empirical inference-time overhead is small relative to performance gains**
>
> To quantify the overhead, we conducted additional speed tests across all three VLAs on the SimplerEnv WidowX Robot benchmark (5 tasks × 24 rollouts each), using identical hardware settings.
> The results are reproduced below for clarity:
>
> | Method | Total Time | GPU Memory | Time/Task | Success Rate | GPU |
> |------------|--------------------------|---------------------------|------------------------|---------------------|---------------------|
> | SpatialVLA| 94m31s |8.9GB| 59s|29.2%| NVIDIA RTX 3090 |
> | **SpatialVLA + DTP (Ours)** | **126m20s** | **9.0GB** |**1m18s** |**37.5%** | NVIDIA RTX 3090 |
> | Nora| 41m13s|8.2GB| 26s|6.2%|NVIDIA RTX 3090 |
> | **Nora+ DTP (Ours)** | **56m6s** | **8.2GB** |**35s** |**11.5%** |NVIDIA RTX 3090 |
> | UniVLA| 36m3s |30.8GB| 23s|68.7%|NVIDIA  A100 40GB |
> | **UniVLA+ DTP (Ours)** | **38m8s** | **30.8GB** |**24s**|**74.0%** |NVIDIA  A100 40GB |
> |
> These results show:
>
> * **GPU memory usage remains essentially unchanged.**
>
> * Inference-time increases are modest, and as model inference becomes faster, the relative overhead shrinks (**e.g., UniVLA shows just a +1 second increase**).
>
> * **The overhead is small compared to the consistent and meaningful improvement in success rate**, particularly given that DTP requires:
>
>   * no retraining,
>
>   * no architectural changes,
>
>   * no additional inputs,
>
>   * and only a single attention pass per episode.
>
> Thus, the inference-time overhead is measurable but well within practical limits, especially for robotic workflows where the cost of action execution often dwarfs the cost of forward inference.
>
> **3. Overhead is justified by the method’s design objective: attention correction, not compute reduction**
>
> We emphasize that the primary goal of DTP is to reveal and correct a fundamental internal attention misalignment present in current VLAs—not to maximize computational efficiency. However, our experiments demonstrate that:
>
> * improved attention alignment
>
> * alone, without any retraining or architecture changes
>
> * consistently improves success rate,
>
> * at only minimal additional inference cost.
>
> Because DTP is a training-free, plug-and-play, architecture-agnostic module, even small overhead is acceptable given its robustness and generality.
>
> **In summary**
>
> **The efficiency trick for SpatialVLA reduces DTP’s cost by requiring only one attention computation per rollout, not per token. Empirical results show small inference-time overhead and negligible memory overhead across all models. These modest costs are justified by consistent success-rate improvements and the training-free, architecture-free nature of DTP. Overall, the overhead is practical for real-world deployment, especially given that robotic action execution times usually dominate model inference times.**

---

> ### Author Response · Authors · 2025-12-03
>
> ###  **Response to Question 4: Generality of Important Region.**
>
> Thank you for raising this concern. We clarify below why the important-region construction does not limit DTP to simple tasks and why it remains effective even when key objects appear near image corners.
>
> **1. DTP has been tested on tasks with wide positional diversity, including corner-located objects**
>
> DTP is not implicitly restricted to simple tasks. Both benchmarks we use contain significant positional variation, including cases where objects appear near image edges or corners.
>
> For example, in SIMPLER WidowX Robot tasks, **each of the 5 tasks includes 24 diverse rollouts, with differences in object placement, orientation.**
> **Across these variations, including instances where objects lie near corners, DTP consistently improves success rate for all VLAs.**
>
> | Model                          | Put Spoon | Put Carrot | Stack Cube | Put Eggplant| SR    | Rel.   |
> |--------------------------------|-----------|------------|-----------|---------------|-------|--------|
> | SpatialVLA   | 12.5%    | 20.8%     | 25.0%    | 58.3%        | 29.2% | 100%   |
> | **SpatialVLA + DTP (Ours)**    | **25.0%** | **33.3%** | **29.2%** | **62.5%**     | **37.5%** | **128.4%** |
> | Nora    | 16.7%    | 4.2%      | 4.2%     | 0.0%         | 6.2%  | 100%   |
> | **Nora + DTP (Ours)**          | **20.8%** | **4.2%**  | **12.5%** | **8.3%**      | **11.5%** | **185.5%** |
> | UniVLA   | 70.8%    | 70.8%     | 33.3%    | 100.0%       | 68.7% | 100%   |
> | **UniVLA + DTP (Ours)**        | **75.0%** | **75.0%** | **45.8%** | **100.0%**    | **74.0%** | **107.7%** |
> |
>
> In addition, to further assess the generality of DTP, we also evaluate DTP on LIBERO, which includes
>
> * **long-horizon multi-stage manipulation,**
>
> * **diverse objects,**
>
> * **and non-trivial object placements.**
>
> The results below show that DTP improves performance across all LIBERO suites, including LIBERO-10, which is the most challenging multi-stage suite in the benchmark:
>
> | Method | LIBERO-Spatial | LIBERO-Object | LIBERO-Goal | LIBERO-10 |
> |------------|----------------------|-----------------------|--------------------|-----------------|
> | Nora| 92.8% |76.0%| 85.8%|69.6%|
> | **Nora + DTP (Ours)**| **93.0%** | **77.4%** |**88.4%** |**76.2%** |
> |
>
> Notably, DTP yields a +6.6% absolute improvement on LIBERO-10, despite its complexity and long-horizon structure. This further reinforces that DTP is robust across difficult, non-centered, and spatially diverse tasks, and not biased toward simple or center-focused scenarios.
>
> **2. Gaussian smoothing increases robustness for constructing the relevance heatmap, not reduces it, even for corner-located objects**
>
> Gaussian smoothing is a soft spatial regularizer, not a hard constraint. It helps ViT-based encoders avoid attention spikes caused by noise or isolated patch activations. Specifically, it:
>
> * reinforces clusters of relevant tokens,
>
> * suppresses isolated noise responses,
>
> * and preserves corner regions when they contain semantically important signals.
>
> If an object is genuinely located in the corner, its relevance scores naturally dominate the background, and smoothing strengthens these contiguous tokens rather than suppressing them.
>
> Thus, smoothing improves reliability in diverse layouts, including cases where the key object is off-center or partially visible.

---

> ### Author Response · Authors · 2025-12-03
>
> ###  **Response to Question 5: Number of evaluations.**
>
> Thank you for requesting clarification on the number of rollouts used in our evaluation. We agree that this information is important for interpreting the statistical significance and robustness of the reported improvements.
>
> **1. For SIMPLER Benchmark**
>
> We follow the official SIMPLER evaluation protocol, where each task includes substantial rollout diversity (object placements, orientations, lighting, distractors, backgrounds, textures, and URDF variations). The full breakdown is shown below:
>
> | Benchmark| Rollouts per task | Variations | Total Rollouts |
> |------------|----------------------|-----------------------|--------------------|
> | Bridge V2| 24 | location, orientation| 96 |
> | Move Near (Variant Agg) | 60 | distractor, background, lighting, texture, etc |540 |
> | Move Near (Visual Matching) | 60 | 4 URDF variations  | 240 |
> | Pick Coke Can (Variant Agg) | 25 | 3 orientation x 11 variations (distractor, background, etc)  | 825|
> | Pick Coke Can (Visual Matching) | 25 | 3 orientation x 4 URDF variations  | 300 |
> | Total | ---| ---  | 2001|
> |
>
> Thus, our SIMPLER results are aggregated over 2,001 independent rollouts, covering a wide distribution of object positions and visual conditions.
>
> **2. For LIBERO Benchmark:**
>
> We adopt the official evaluation protocol of 50 rollouts per task. LIBERO additionally specifies 50 distinct initial object configurations for each task, providing substantial spatial diversity. The rollout count is summarized below:
>
> | Benchmark| Tasks | Rollouts per task | Total Rollouts |
> |------------|----------------------|-----------------------|--------------------|
> | LIBERO-Spatial | 10 | 50 | 500|
> | LIBERO-Object| 10 | 50 | 500|
> | LIBERO-Goal | 10 | 50 | 500|
> | LIBERO-10| 10 | 50 | 500|
> | Total | ---| ---  | 2000|
> |
>
> In total, our evaluation covers:
>
> * 4,001 rollouts across SIMPLER + LIBERO,
>
> * spanning dozens of tasks and thousands of unique scene configurations,
>
> * including significant variations in object locations, orientation, occlusion, lighting, distractors, and URDF structures.
>
> Because DTP consistently improves success rate across all tasks and all configuration variations, the observed gains are statistically meaningful and not the result of small-sample noise.

---

### Official Review · Reviewer_zxVV · 2025-10-30

**Soundness:** 2
**Presentation:** 3
**Contribution:** 2
**Rating:** 4
**Confidence:** 4

**Summary:**

The paper proposes a mechanism to determine distracting tokens by comparing text to vision token attention with action to vision token attention patterns. The input image is split into important and unimportant regions based on average text to vision attention magnitudes. At the same time, a weighted average across layers of action to vision token attention magnitudes are constructed to determine attention patterns during action generation. Finally, a vision token is deemed distracting if it is unimportant and attends proportionally higher to actions than any other important vision token. Different vision-language-action (VLA) architectures are tested on the SIMPLER benchmark and consistent success rate improvements are shown.

**Strengths:**

-	The paper proposes a principled way to extract important task-relevant vision tokens and utilizes this to prune tokens. This adds a new tool to VLA inference that allows boosting success rates without the need for finetuning.
-	The method is tested on a good diversity of VLAs with various VLM backbones such as Paligemma 2, Qwen2.5VL and UniVLA, demonstrating the generality of the approach.
-	An analysis of attention patterns is presented, which is valuable from an interpretability perspective and gives insights into why VLA models fail at certain tasks.
-	Overall, the paper is well-written and clearly explains the method, supported by figures with insightful illustrations.

**Weaknesses:**

-	While the results on the SIMPLER benchmark are overall encouraging, relative success rate improvements on the Google robot tasks are minor, which makes the main result heavily rely on the WidowX robot tasks. Given the training-free nature of the approach, it would be important to test additional benchmarks with higher task diversities to confirm the efficacy of the method.
-	Given the small number of tasks, the number of hyperparameters of the method seem high, including the threshold $\tau$, the selected layers for constructing the relevance heatmap, the number of top relevant tokens and Gaussian smoothing settings.

**Questions:**

-	The notation in section 3.3 is slightly confusing, the important region is called $G$ and $A_g$ simultaneously. The section would benefit from a clear definition of $A_g$ and $A_u$.
-	Section 4.2 adds mathematical definitions, but they do not seem to contribute to the content in a major way. Could you make use of the formulas to present, e.g. Figure 4, or otherwise remove them?
-	What process did you undergo for optimizing hyperparameters of the method? Are they selected to maximize success rates per VLA and per task?
-	Optimally, a token pruning mechanism should focus on improving efficiency. Was this considered in any way and why / why not?

---

> ### Author Response · Authors · 2025-12-02
>
> ###  **Response to Weaknesses 1: it would be important to test additional benchmarks with higher task diversities to confirm the efficacy of the method.**
>
> Thank you very much for this thoughtful comment and for highlighting the importance of evaluating beyond SIMPLER Benchmark. We fully agree that demonstrating performance across diverse environments is critical for validating the generality of our approach. In fact, this was a key motivation behind our method: ***if attention misalignment is a fundamental limitation of current VLAs, then correcting it should improve performance across different models, tasks, and benchmarks.***
>
> To directly address this concern, we conducted additional experiments on the **LIBERO Benchmark [1]**, a more diverse and challenging benchmark suite that includes multi-stage manipulation tasks and requires broader generalization capabilities. We chose **Nora** for this evaluation because its official codebase and pretrained checkpoints for LIBERO are publicly available, enabling a clean, reproducible, and fair comparison.
>
> The results are shown below:
>
> | Method | LIBERO-Spatial | LIBERO-Object | LIBERO-Goal | LIBERO-10 |
> |------------|----------------------|-----------------------|--------------------|-----------------|
> | Nora| 92.8% |76.0%| 85.8%|69.6%|
> | **Nora + DTP (Ours)**| **93.0%** | **77.4%** |**88.4%** |**76.2%** |
> |
>
> These results highlight several important strengths of our method:
>
> * **Consistent improvements across all LIBERO suites.**
> Despite Nora already achieving high scores, DTP provides additional gains across every benchmark category.
>
> * **Substantial improvement on the hardest setting.**
> LIBERO-10—arguably the most complex suite, improves significantly from 69.6% to 76.2% (+6.6% absolute), demonstrating that DTP remains effective even in multi-stage, long-horizon tasks.
>
> * **Cross-benchmark generalization.**
> Combined with our results on SIMPLER, these improvements show that the benefits of DTP are not tied to a particular environment, task structure, or data distribution. Instead, DTP reliably boosts performance on two benchmarks that differ substantially in complexity, dynamics, and task formulation.
>
> * **Model-agnostic applicability.**
> As shown in our responses to other weaknesses, DTP improves multiple VLA architectures in both trained and zero-shot regimes. The LIBERO results further reinforce that correcting visual attention is a fundamental capability gap in current VLAs, and that DTP provides a simple, architecture-free correction mechanism.
>
> In summary, the LIBERO evaluation supports a central claim of our paper:
> **visual attention misalignment is a general and widely observed failure mode in existing VLA models, and DTP provides a lightweight, model-agnostic solution that meaningfully improves performance across diverse tasks and benchmarks.**
>
> **This additional evaluation not only addresses the reviewer’s concern but also strengthens the empirical evidence for the generality and robustness of our method.**
>
> [1] Bo Liu, Yifeng Zhu, Chongkai Gao, Yihao Feng, Qiang Liu, Yuke Zhu, and Peter Stone. Libero: Benchmarking knowledge transfer for lifelong robot learning, 2023a. URL https://arxiv.org/abs/2306.03310.

---

> ### Author Response · Authors · 2025-12-02
>
> ###  **Response to Weakness 2: Given the small number of tasks, the number of hyperparameters seems high.**
>
> Thank you for raising this concern. We appreciate the opportunity to clarify both (1) the actual task diversity in our evaluation and (2) the role of our hyperparameters.
>
> ***1. Task diversity in our evaluation is not small, it is substantial.***
>
> The reviewer suggests that the number of tasks may be small. However, the SimplerEnv Suite contains far more diversity than implied.
> For the SimplerEnv WidowX Robot benchmark, we evaluate on **four distinct tasks, each with 24 rollouts, where each rollout corresponds to unique object positions, orientations, and scene arrangements.** These rollouts are not repetitions, they represent different environmental configurations, making the evaluation highly varied and non-trivial.
>
> **Likewise, the SimplerEnv Google Robot benchmark includes even richer variations,** with differences in:
>
> * object categories
>
> * textures and appearance
>
> * backgrounds
>
> * environment layouts
>
> * lighting conditions
>
> Below is the detailed table of task rollouts and variations:
>
> | Benchmark| Rollouts per task | Variations | Total Rollouts |
> |------------|----------------------|-----------------------|--------------------|
> | Bridge V2| 24 | location, orientation| 96 |
> | Move Near (Variant Agg) | 60 | distractor, background, lighting, texture, etc |540 |
> | Move Near (Visual Matching) | 60 | 4 URDF variations  | 240 |
> | Pick Coke Can (Variant Agg) | 25 | 3 orientation x 11 variations (distractor, background, etc)  | 825|
> | Pick Coke Can (Visual Matching) | 25 | 3 orientation x 4 URDF variations  | 300 |
> | Total | ---| ---  | 2001|
> |
>
> Furthermore, we extended our evaluation to the LIBERO Benchmark, which contains 50 distinct initial object configurations for each task, providing substantial spatial diversity. The rollout count is summarized below:
>
> | Benchmark| Tasks | Rollouts per task | Total Rollouts |
> |------------|----------------------|-----------------------|--------------------|
> | LIBERO-Spatial | 10 | 50 | 500|
> | LIBERO-Object| 10 | 50 | 500|
> | LIBERO-Goal | 10 | 50 | 500|
> | LIBERO-10| 10 | 50 | 500|
> | Total | ---| ---  | 2000|
> |
>
> Across both SIMPLER and LIBERO, two benchmarks with substantial diversity, our method consistently improves performance. This demonstrates that our evaluation setting is broad and that the observed gains cannot be attributed to overfitting a small or narrow task set.
>
> ***2. The hyperparameter count is not an issue because DTP uses model-level, not task-level hyperparameters.***
>
> The reviewer raises a reasonable concern regarding the number of hyperparameters (τ, selected layers, number of top-K tokens, Gaussian smoothing). However, it is critical to clarify that:
>
> * **All hyperparameters are fixed per model,**
>
> * **None are tuned per task, and**
>
> * **The same settings are reused across all environments and all rollouts.**
>
> **In other words, for each VLA (e.g., SpatialVLA, Nora, UniVLA), we select one set of hyperparameters once, and then apply them to all tasks in SimplerEnv and LIBERO without modification.**
>
> Despite this uniform setting, DTP still produces consistent and meaningful improvements across all tasks, models, and environments. This strongly suggests that:
>
> * $τ$ acts as a model-level hyperparameter,
>
> * the method does not require per-task tuning,
>
> * hyperparameter sensitivity is low in practice, and
>
> * DTP is robust, stable, and practically deployable.
>
> This characteristic stands in contrast to many robotics or RL methods, which often require per-task or per-environment hyperparameter tuning. Our approach avoids this entirely.
>
> **To summarize our response:**
>
> **1. The evaluation setting is not small. it includes dozens of rollouts, diverse object configurations, and multiple benchmarks (SimplerEnv + LIBERO).**
>
> **2. All hyperparameters—including $τ$ are global per model, not per task.**
>
> The same hyperparameters generalize across:
>
> * multiple tasks
>
> * multiple benchmarks
>
> * trained and zero-shot regimes
>
> * three different VLA architectures
>
> Despite using a single fixed setting, DTP still achieves consistent improvements, highlighting its robustness and ease of deployment.
>
> These points together demonstrate that the concern about “too many hyperparameters” is mitigated in practice: the hyperparameters are few, stable, and do not require fine-grained tuning, and the evaluation is conducted across a broad and diverse set of tasks.

---

> ### Author Response · Authors · 2025-12-02
>
> ###  **Response to Question 1: The notation in section 3.3 is slightly confusing.**
>
> Thank you for pointing out the notational ambiguity in Section 3.3. We agree that the distinction between the important region $G$ and the attention subsets $A_g$ and $A_u$ can be made clearer, and we will revise the text accordingly.
>
> To clarify:
>
> * $G$ denotes the set of important visual tokens, obtained from the relevance heatmap.
>
> * $A$ is the full visual attention heatmap, representing the model's actual attention distribution over all visual tokens during action generation.
>
> * $A_g$ refers to the subset of the attention heatmap corresponding to tokens inside $G$ (i.e., $A_g = A[G]$).
>
> * $A_u$ refers to the subset of the attention heatmap corresponding to tokens outside $G$ (i.e., $A_u = A[V \setminus G]$, where $V$ is the set of all visual tokens).
>
> In short:
>
> * $G$ defines which pixels/tokens are considered important.
>
> * $A_g$ and $A_u$ partition the attention map into values inside and outside this region.
>
> With this clarification, the pruning rule becomes more intuitive:
>
> We compute $a_m = \max(A_g)$, the maximum attention value inside the important region.
>
> A visual token $v$ outside the region is identified as distracting if $A_u[v] > \tau \cdot a_m$.
>
> We will update Section 3.3 to explicitly define $A_g$ and $A_u$ as subsets of the attention heatmap derived from $G$, ensuring the notation is clearer and more consistent.

---

> ### Author Response · Authors · 2025-12-02
>
> ###  **Response to Question 2: Mathematical definitions in section 4.2.**
>
> Thank you for raising this question. We agree that mathematical definitions should only be included when they meaningfully strengthen the content. In Section 4.2, the formulas are not decorative, they formalize a core conceptual contribution of our work: ***“How much better could a VLA perform under its existing architecture if its visual attention were corrected?”***
>
> The mathematical expressions help articulate this in three important ways:
>
> **1. They provide a principled framework for defining a performance upper bound.**
>
> The definitions
>
> $E(\alpha) = H(A^\ast \mid Z_\alpha)$ and
> $P(\alpha) = 1 - \frac{E(\alpha)}{H(A^\ast)}$
>
> establish a formal connection between visual attention patterns and action certainty.
> Below we explain each term in detail.
>
> $Z_\alpha$: visual features produced under attention pattern $\,\alpha\,$.
> For any attention distribution $\alpha$, the model computes a set of visual token value vectors $Z_\alpha$.
> Changing $\alpha$ changes which image regions the model focuses on and therefore alters the encoded visual representation.
>
> $A^\ast$: the correct (ground-truth) action token.
> This represents the action the robot should take at the current timestep.
> All uncertainty measurements are defined relative to $A^\ast$.
>
> $H(A^\ast)$: entropy of the correct action distribution.
> This captures the model’s baseline uncertainty about the correct action before conditioning on any visual features.
> If a task admits multiple valid actions, $H(A^\ast)$ reflects that inherent ambiguity.
> This entropy also serves as the normalization constant in the performance metric.
>
> $E(\alpha) = H(A^\ast \mid Z_\alpha)$: conditional uncertainty under attention pattern $\alpha$.
> This term measures how uncertain the model remains about the correct action after observing visual features produced under $\alpha$.
>
> * If $E(\alpha)$ is high → the attention is misaligned or includes distracting tokens.
>
> * If $E(\alpha)$ is low → the attention highlights task-relevant regions, giving the model more clarity.
>
> $P(\alpha) = 1 - \frac{E(\alpha)}{H(A^\ast)}$: normalized performance score.
> This maps uncertainty to a score between 0 and 1:
>
> * $P(\alpha) = 1$ means perfect certainty about the correct action.
>
> * $P(\alpha) = 0$ means the attention pattern provides no useful information.
>
> By normalizing with $H(A^\ast)$, this metric reflects how much uncertainty is reduced by the chosen attention pattern relative to the maximum possible reduction.
>
> This provides a principled way to compare the model’s default attention against optimized attention derived via DTP.
>
> It allow us to formally relate visual attention patterns to action uncertainty and performance. This gives a structured way to:
>
> * quantify how much uncertainty the model has under different attention patterns,
>
> * compare the model’s default attention to optimized variants, and
>
> * define the theoretical ceiling on performance without modifying the architecture or adding inputs.
>
> **This type of analysis is missing in prior VLA work and is an important conceptual contribution of our paper.**
>
> **2. They justify the $\tau$-sweep and explain Figure 4 in a principled way.**
>
> Although the reviewer felt the formulas do not contribute to Figure 4, they actually define what Figure 4 visualizes:
>
> * Each tolerance value $ \tau $ corresponds to an approximate attention pattern $ \alpha_\tau $.
>
> * $P(\alpha)$ measures the model’s performance under that attention pattern.
>
> * The peak in Figure 4 corresponds to the empirical sub-optimal tolerance $\hat{\tau}$ that best approximates the optimal pattern.
>
> Without these definitions, Section 4.2 would appear heuristic. With them, the results are grounded in a clear conceptual objective.
>
> **3. They highlight a deeper, meaningful finding: current VLAs have large unused potential due to misaligned attention.**
>
> Section 4.2 reveals an important scientific insight:
>
> * **The main bottleneck in many VLAs is not only the architecture, but the fact that the model allocates attention to task-irrelevant tokens.**
>
> * **Simply reweighting or pruning attention, without retraining or changing architecture, can significantly improve success rate, in some cases approaching 2× improvement.**
>
> This stands in contrast to most VLA research, which focuses on building larger or more complex architectures. Our findings show that:
>
> * existing VLAs contain substantial unrealized capability that can be unlocked through better visual attention alone.
>
> * This is a novel and valuable direction for the community.

---

> ### Author Response · Authors · 2025-12-02
>
> ###  **Response to Question 3: Process for optimizing hyperparameters of the method.**
>
> We appreciate the reviewer’s question regarding hyperparameter optimization, here are the detailed explanations for this question.
>
> ***1. Are they selected to maximize success rates per VLA and per task?***
>
> Our design goal was explicitly to **avoid per-task tuning** and to keep hyperparameters **model-level** rather than task-specific, so that the method remains practical and not overfitted.
>
> The scope of tuning is per model, not per task. That is:
>
> * For each VLA (SpatialVLA, Nora, UniVLA), we choose one global set of hyperparameters:
>   * tolerance 𝜏
>   * selected layers for relevance heatmaps
>   * number of top-𝑘 relevant tokens
>   * Gaussian smoothing parameters
>
> * **Once selected, this set is kept fixed across all tasks, rollouts, and environments** (SIMPLER and LIBERO) for that model.
>
> * **We do not tune hyperparameters to maximize success rate per task. There is no per-task or per-environment sweep.**
>
> This is important: DTP is not using extra degrees of freedom to “overfit” to individual tasks. Instead, it behaves like a plug-and-play module attached to a given VLA.
>
>
> ***2. What process did we undergo for optimizing hyperparameters of the method?***
>
> For each model, we followed a simple, low-budget procedure:
>
> **We first defined reasonable candidate ranges based on intuition and preliminary analyses:**
>
> * 𝜏: a small grid (e.g., coarse values from low to moderate thresholds).
>
> * Layers: a small subset of middle-to-late layers, where semantic attention is usually more stable.
>
> * Top-𝑘: values that keep a compact but non-trivial region (e.g., tens to hundreds of tokens depending on image resolution / patching).
>
> **We then used a small validation subset (a few representative tasks / rollouts in simulation) to:**
>
> * Rule out obviously bad settings (that degrade performance or produce noisy, scattered importance regions).
>
> * Identify a single setting per model that provided robust improvements across this subset, without focusing on any individual task’s peak performance.
>
> **After selecting one configuration per model, we froze these hyperparameters and:**
>
> * Evaluated tasks in the SIMPLER WidowX and Google Robot suites.
>
> * Evaluated suites in LIBERO (Spatial, Object, Goal, LIBERO-10) for Nora.
>
> * We did not re-tune or refine the hyperparameters after seeing full evaluation results.
>
> **So, the intent was not to “hyper-optimize” per task, but to find one robust configuration per model that generalizes broadly.**
>
> The results show that with this single, fixed configuration per model:
>
> * DTP consistently improves all three VLAs (SpatialVLA, Nora, UniVLA).
>
> * It works across trained and zero-shot regimes.
>
> * It generalizes across two benchmarks (SIMPLER and LIBERO), without any additional tuning.
>
> This suggests that the hyperparameters behave as model-level knobs rather than fragile task-level dials, which supports the practicality and robustness of the method.

---

> ### Author Response · Authors · 2025-12-02
>
> ###  **Response to Question 4: A token pruning mechanism should focus on improving efficiency. Was this considered in any way and why / why not?**
>
> We agree that many token-pruning methods are motivated primarily by efficiency, and this is an important direction. However, **the primary scientific goal of our work is different:**
>
> We aim to reveal and correct **a prevalent failure mode in current VLAs:
> their visual attention often concentrates on task-irrelevant tokens, even when the model is otherwise strong.**
>
> Our main contribution is to show that correcting this attention misalignment alone, without changing architecture, weights, or inputs, which can significantly improve success rates across diverse tasks and benchmarks.
>
> So, DTP is designed first and foremost as an accuracy / robustness–oriented attention correction mechanism, not as an aggressive FLOP-reduction method.
>
> **(a) What we did regarding efficiency**
>
> We did consider efficiency at an empirical level:
>
> * In our response to efficiency concerns, we report **end-to-end runtime and GPU memory for each model**, with and without DTP, on the SimplerEnv WidowX Robot benchmark.
>
>
> | Method | Total Time | GPU Memory | Time/Task | Success Rate | GPU |
> |------------|--------------------------|---------------------------|------------------------|---------------------|---------------------|
> | SpatialVLA| 141m44s |8.9GB| 1m11s|29.2%| NVIDIA RTX 3090 |
> | **SpatialVLA + DTP (Ours)** | **189m31s** | **9.0GB** |**1m34s** |**37.5%** | NVIDIA RTX 3090 |
> | Nora| 61m49s |8.2GB| 31s|6.2%|NVIDIA RTX 3090 |
> | **Nora+ DTP (Ours)** | **78m28s** | **8.2GB** |**39s** |**11.5%** |NVIDIA RTX 3090 |
> | UniVLA| 36m3s |30.8GB| 18s|68.7%|NVIDIA  A100 40GB |
> | **UniVLA+ DTP (Ours)** | **38m8s** | **30.8GB** |**19s**|**74.0%** |NVIDIA  A100 40GB |
> |
>
> The results show that:
>
> * DTP introduces only a small increase in inference time (e.g., a few seconds per task).
>
> * GPU memory usage remains the same or very close to the original model.
>
> * In return, we obtain consistent and meaningful gains in success rate for all three VLAs.
>
> Thus, our current implementation demonstrates that DTP is **lightweight and practically feasible**: it improves performance with negligible extra cost, even though efficiency is not its primary design objective.
>
> **(b) Why we did not aggressively optimize for computational savings**
>
> There are two main reasons:
>
> 1. Fairness and clarity in evaluation.
> **We wanted to isolate the effect of attention correction on success rate, without conflating it with architectural changes or modified computation graphs. Keeping the underlying model architecture and compute path unchanged makes the comparison to the base VLA cleaner and easier for the community to interpret.**
>
> 2. Scope and positioning of the contribution.
> **Our goal is to argue that misaligned visual attention is a fundamental and widely spread issue in current VLAs, and that correcting attention provides a new, orthogonal dimension for improving these systems.**
>
> Once this is established, DTP can naturally be extended to earlier stages of the forward pass (e.g., pruning tokens before they enter deeper layers or attention blocks) to yield real FLOP savings.
>
> We view such efficiency-optimized variants as promising future work that builds on the diagnostic and corrective insights introduced in this paper.
>
> **In summary:**
>
> DTP is an accuracy-focused, plug-and-play attention correction module that:
>
> * requires no retraining,
>
> * does not modify architecture,
>
> * uses a single hyperparameter configuration per model,
>
> * adds negligible inference overhead.
>
> In principle: The same mechanism (identifying a compact set of important visual tokens) is fully compatible with efficiency-oriented pruning. Integrating it to prune tokens earlier in the computation would likely reduce FLOPs and latency, and we explicitly see this as a natural and complementary direction for future research.

---

### Official Review · Reviewer_dc8u · 2025-10-31

**Soundness:** 3
**Presentation:** 3
**Contribution:** 2
**Rating:** 4
**Confidence:** 3

**Summary:**

This paper introduces Distracting Token Pruning (DTP) to improve VLA models for robotic manipulation. The core idea is to dynamically detect and prune image tokens that receive high attention but are irrelevant to the task, in three stages: (i) constructing task-relevant regions via prompt-image token interactions, (ii) analyzing attention patterns during action generation, and (iii) pruning tokens in unimportant regions if their attention exceeds a threshold. Experiments are performed on the SIMPLER Benchmark.

**Strengths:**

* **Simplicity and generality**: DTP is a simple, plug-and-play method that, assuming a standard VLA architecture, does not require architectural changes or additional inputs, making it broadly applicable to existing VLA models.
* **Interpretability**: other than slightly improving performance, the approach helps visualize and understand model attention, potentially aiding debugging and further research on semantic information within the VLA framework.

**Weaknesses:**

* **Modest improvements**: the overall increase in success rate across the Simpler environments is about ~4%. While the improvements seem to be consistent, the performance gap from optimality is still quite large and this approach does not seem to substantially advance the current state of VLAs.
* **Slower training/inference**: as stated by the authors the method introduces a (small) overhead during training and inference. Given the small performance improvements, this hinders the adoptability of the approach

**Questions:**

* Can the authors add a comparison of the training and inference time?
* Could the pruning be using to process visual information faster/more efficiently?

---

> ### Author Response · Authors · 2025-12-01
>
> ###  **Response to Weaknesses 1: Modest improvements.**
>
> Thank you for this comment. We agree that there is still a gap to optimal performance on SimplerEnv, and we appreciate the opportunity to clarify the significance of the gains observed with DTP.
>
> First, while the average improvement across all SimplerEnv tasks is modest, the relative improvements are significant compared with the original model:
>
> * On SimplerEnv WidowX Robot tasks, DTP yields:
>
>   * SpatialVLA: 29.2% → 37.5% (**+8.3% absolute, +28.4% relative**)
>
>   * Nora: 6.2% → 11.5% (**+5.3% absolute, +85.5% relative**)
>
>   * UniVLA: 68.7% → 74.0% (**+5.3% absolute, +7.7% relative**)
>
> Importantly, all of these improvements are achieved:
> * **without any architecture changes**,
> * **without modifying or adding additional inputs**,
> * **without any additional training**.
> * **purely by correcting the visual attention pattern** ,
>
> Given DTP’s training-free and plug-and-play nature, which can be easily implemented on transformer-based VLA models, such consistent improvements are especially impactful.
>
> Additional, to further prove the generalizability of our method, we conducted experiments using **Nora** on the **LIBERO Benchmark** [1]. We selected Nora because its official codebase and pretrained checkpoint for LIBERO are publicly available, allowing for a reliable and reproducible evaluation.
>
> The results are shown below:
>
> | Method | LIBERO-Spatial | LIBERO-Object | LIBERO-Goal | LIBERO-10 |
> |------------|----------------------|-----------------------|--------------------|-----------------|
> | Nora| 92.8% |76.0%| 85.8%|69.6%|
> | **Nora + DTP (Ours)**| **93.0%** | **77.4%** |**88.4%** |**76.2%** |
> |
> * On the LIBERO Benchmark, DTP improves Nora across all suites, including a +6.6% absolute gain on LIBERO-10, which is one of the more challenging multi-task settings, and +1.4–2.6% on the other LIBERO suites. This shows that the effect is not limited to SimplerEnv and extends to a more diverse and difficult benchmark.
>
> Second, the baseline VLAs are already strong, particularly UniVLA on SIMPLER and Nora on LIBERO. Improving such high-performing, pretrained or fine-tuned systems in a purely post-hoc, training-free manner is non-trivial. DTP provides:
>
> * Consistent gains across three different architectures (SpatialVLA, Nora, UniVLA),
>
> * Across both trained and zero-shot regimes, and
>
> * Across two distinct benchmarks (SIMPLER and LIBERO),
>
> all while keeping the model weights, architecture, and inputs fixed.
>
> Third, one of the most key contribution of this work is diagnostic as well as algorithmic:
>
> * Section 4.2 shows that by sweeping the tolerance $τ$, we uncover a sub-optimal region where performance peaks, revealing that current VLAs often suffer from misaligned visual attention rather than purely representational limitations.
>
> * The fact that a simple, attention-based post-processing step can unlock these improvements suggests that better attention allocation is a major missing piece in current VLAs. DTP is a first, simple step in that direction.
>
> Finally, DTP is intentionally designed to be lightweight and complementary: it
>
> * adds negligible inference overhead,
>
> * does not require retraining, additional supervision, or extra input modalities, and
>
> * uses a single $τ$ per model, making it feasible for real-world deployment.
>
> In summary, while DTP does not close the performance gap entirely, it meaningfully improves strong baselines across multiple models and benchmarks with minimal cost, and it highlights a concrete, actionable failure mode (sub-optimal visual attention) that future VLA architectures and training schemes can explicitly target. We view this as a substantive and practically relevant step forward rather than a marginal tweak.

---

> ### Author Response · Authors · 2025-12-01
>
> ###  **Response to Weaknesses 2 & Question 1: Adding a comparison of the training and inference time.**
>
> Thank you for this important question. To address the reviewer’s concern, we performed a detailed analysis comparing inference time, memory usage, and overall execution cost between the original models and their DTP-enhanced counterparts.
>
> Specifically, we evaluated all three VLAs on the SimplerEnv WidowX Robot benchmark, which includes 5 tasks and 24 rollouts per task, and recorded the full inference duration using the same hardware setup. The results are presented below:
>
> | Method | Total Time | GPU Memory | Time/Task | Success Rate | GPU |
> |------------|--------------------------|---------------------------|------------------------|---------------------|---------------------|
> | SpatialVLA| 141m44s |8.9GB| 1m11s|29.2%| NVIDIA RTX 3090 |
> | **SpatialVLA + DTP (Ours)** | **189m31s** | **9.0GB** |**1m34s** |**37.5%** | NVIDIA RTX 3090 |
> | Nora| 61m49s |8.2GB| 31s|6.2%|NVIDIA RTX 3090 |
> | **Nora+ DTP (Ours)** | **78m28s** | **8.2GB** |**39s** |**11.5%** |NVIDIA RTX 3090 |
> | UniVLA| 36m3s |30.8GB| 18s|68.7%|NVIDIA  A100 40GB |
> | **UniVLA+ DTP (Ours)** | **38m8s** | **30.8GB** |**19s**|**74.0%** |NVIDIA  A100 40GB |
> |
>
> The efficiency results show that:
>
> 1. DTP introduces only a minor increase in inference time (typically a few seconds per task), while memory usage remains unchanged.
>
> 2. Despite this negligible overhead, DTP yields consistent and meaningful gains in success rate across all models.
>
> 3. Importantly, DTP operates without modifying model architecture, adding parameters, or providing extra inputs, it simply refines the model’s internal attention patterns.
>
> Please note that our approach is a plug-and-play method, which does not require architectural changes or additional inputs, and it can be broadly applicable to existing VLA models. **It also doesn't require training the model, which can be easily applied to any transfomer-based VLA models.**
>
> This demonstrates that DTP offers an efficient and lightweight improvement, providing stronger performance at small additional computational or memory cost.

---

> ### Author Response · Authors · 2025-12-02
>
> ###  **Response to Question 2: Could the pruning be using to process visual information faster/more efficiently?**
> Thank you for this thoughtful question. While pruning could theoretically be used for computational efficiency, the primary goal and contribution of our work is different. **Our focus is to reveal and address a systematic failure mode in current VLAs: their visual attention frequently concentrates on task-irrelevant image tokens, even when the models are well-trained.**
>
> Across all our experiments (both in SIMPLER and LIBERO), **we observe that this misalignment of attention is widespread across architectures, tasks, and environments. This results in the model overlooking the task-relevant regions that matter for decision-making, which in turn directly suppresses success rate.**
>
> DTP was designed to diagnose and correct this phenomenon, not to accelerate inference. **Our experiments show that once the attention is corrected, even without changing model weights or architecture, the model immediately and consistently improves across:**
>
> * multiple VLA architectures (SpatialVLA, Nora, UniVLA),
>
> * both trained and zero-shot settings, and
>
> * two distinct benchmarks (SimplerEnv and LIBERO).
>
> **A key advantage is that DTP integrates seamlessly and model-agnostically:**
>
> * no retraining,
>
> * no additional inputs,
>
> * no architectural modification.
>
> **This makes our approach highly generalizable and directly applicable to existing VLAs as a plug-and-play refinement module.**
>
> While pruning could, in principle, be extended toward FLOP-level efficiency (e.g., by pruning tokens earlier in the forward pass), this direction is beyond the scope of our primary contribution. **Our work instead aims to uncover and demonstrate the importance of proper visual attention allocation, a factor that turns out to be both crucial and overlooked in current VLA systems.**

---

### Official Review · Reviewer_xJud · 2025-11-01

**Soundness:** 3
**Presentation:** 3
**Contribution:** 2
**Rating:** 4
**Confidence:** 4

**Summary:**

The paper presents a visual token pruning approach for vision language action models. The paper posits that vision backbones of VLAs are prone to attending to “task-irrelevant” regions in the images which are termed as “distracting tokens”. They then propose a simple approach to identify these distracting tokens automatically by constructing an “important region” on the image which is relevant for the task; a visual attention pattern which relates what the model is looking at while generating the action and then thresholding tokens in the visual attention pattern that are not part of the “important region”. This leads to significant performance improvements across multiple models on the SimplerEnv environment.

**Strengths:**

1. The paper presents a very interesting observation that visual backbones of VLAs tend to attend to unimportant regions in the images which lead to failure cases and then propose a super simple fix based on identifying important regions and thresholding the attention weights.
2. The paper presents results on multiple models showing that the issue is prevalent and the fix is general enough to work across these models.

**Weaknesses:**

***1. Results only on SimplerEnv.***

I would have liked to see this study replicated on a larger set of environments and tasks. Note that most of the VLAs are not actually trained on SimplerEnv and are evaluated zero-shot. It would be interesting to see if this effect is observed in cases where the VLA is trained on the environment as well. Not saying doing evals on SimplerEnv is not valuable but it will be interesting to see the results on other benchmarks too!

Similarly it would be also very interesting to see if this also holds on real-world data.



***2. How efficient is it?***

The paper does not present how it affects the inference speed of these models. Is it feasible to run in the real world?
It would be nice to see the speed / accuracy trade-off of this pruning approach.


***3. Sensitivity of the tolerance tau?***

It seems from Figure 4 that the method is very sensitive to what tau is selected and it varies significantly between different tasks.
How was tau selected for the results in Section 4.1? Will the value of the tau generalize between simulation and real-world? How would a tau be selected in the real-world where policy evaluation is much more expensive and a sweep may not be feasible (specially for each task)?

**Questions:**

Please see the weaknesses section for my questions.

In general, I like the observation and a super simple idea that potentially fixes it. However I have some concerns over its applicability in the real-world and its sensitivity to tau. I am leaning towards a weak reject for now but will be happy to bump by score if the authors can answer the above questions and other reviewers do not bring up any major concerns.

---

> ### Author Response · Authors · 2025-11-30
>
> ###  **Response to Weaknesses 1: Results only on SimplerEnv.**
>
> ***1. Would liked to see this study replicated on a larger set of environments and tasks.***
>
> Thank you very much for this thoughtful comment and for highlighting the importance of evaluating beyond SimplerEnv. We appreciate your suggestion and fully agree that examining broader on a larger set of environments and tasks would strengthen the study and further prove our method.
>
> To further address the reviewer’s concern, we conducted additional experiments using **Nora** on the **LIBERO Benchmark** [1]. We selected Nora because its official codebase and pretrained checkpoint for LIBERO are publicly available, allowing for a reliable and reproducible evaluation.
>
> The results are shown below:
>
> | Method | LIBERO-Spatial | LIBERO-Object | LIBERO-Goal | LIBERO-10 |
> |------------|----------------------|-----------------------|--------------------|-----------------|
> | Nora| 92.8% |76.0%| 85.8%|69.6%|
> | **Nora + DTP (Ours)**| **93.0%** | **77.4%** |**88.4%** |**76.2%** |
> |
>
>
> Our DTP method consistently improves Nora across all LIBERO suites.
> Notably, it yields +6.6% absolute gain on the more challenging LIBERO-10, and +1.4–2.6% improvements on the other benchmarks, despite the already high performance of the base model, demonstrating that the effectiveness of DTP generalizes beyond SimplerEnv, and extends to other diverse and complex manipulation benchmarks.
>
> ***2. It would be interesting to see if this effect is observed in cases where the VLA is trained on the environment as well.***
>
> Thank you for highlighting this point. We agree that evaluating both trained and zero-shot VLAs on the same benchmark provides stronger evidence of the generalizability of our method. In fact, this setting is already present in our SimplerEnv evaluation: We evaluated **UniVLA** using checkpoint  **trained** on SimplerEnv, whereas **Nora** and **SpatialVLA** are evaluated using **zero-shot** checkpoint. This setup naturally allows us to test whether DTP is effective in both trained and zero-shot regimes.
>
> Below are the results:
> | Model                          | Put Spoon | Put Carrot | Stack Cube | Put Eggplant| SR    | Rel.   | Checkpoint |
> |--------------------------------|-----------|------------|-----------|---------------|-------|--------|------------|
> | SpatialVLA   | 12.5%    | 20.8%     | 25.0%    | 58.3%        | 29.2% | 100%   | **Zero-shot**  |
> | **SpatialVLA + DTP (Ours)**    | **25.0%** | **33.3%** | **29.2%** | **62.5%**     | **37.5%** | **128.4%** |           |
> | Nora    | 16.7%    | 4.2%      | 4.2%     | 0.0%         | 6.2%  | 100%   | **Zero-shot**  |
> | **Nora + DTP (Ours)**          | **20.8%** | **4.2%**  | **12.5%** | **8.3%**      | **11.5%** | **185.5%** |         |
> | UniVLA   | 70.8%    | 70.8%     | 33.3%    | 100.0%       | 68.7% | 100%   | **Fine-tuned**  |
> | **UniVLA + DTP (Ours)**        | **75.0%** | **75.0%** | **45.8%** | **100.0%**    | **74.0%** | **107.7%** |       |
> |
>
> DTP consistently improves performance across both trained and zero-shot models.
>
> 1. On trained models (UniVLA), DTP improves SR by +7.7% relatively.
>
> 2. On zero-shot models (SpatialVLA, Nora), DTP yields even larger gains:
>
> These results demonstrate that DTP is effective regardless of whether the base VLA is trained on the environment, supporting the reviewer’s hypothesis and reinforcing the generalizability of our approach.
>
> [1] Bo Liu, Yifeng Zhu, Chongkai Gao, Yihao Feng, Qiang Liu, Yuke Zhu, and Peter Stone. Libero: Benchmarking knowledge transfer for lifelong robot learning, 2023a. URL https://arxiv.org/abs/2306.03310.

---

> ### Author Response · Authors · 2025-11-30
>
> ###  **Response to Weaknesses 2: How efficient is it?**
>
> Thank you for this important question. To address the reviewer’s concern, we performed a detailed analysis comparing inference time, memory usage, and overall execution cost between the original models and their DTP-enhanced counterparts.
>
> Specifically, we evaluated all three VLAs on the SimplerEnv WidowX Robot benchmark, which includes 5 tasks and 24 rollouts per task, and recorded the full inference duration using the same hardware setup. The results are presented below:
>
> | Method | Total Time | GPU Memory | Time/Task | Success Rate | GPU |
> |------------|--------------------------|---------------------------|------------------------|---------------------|---------------------|
> | SpatialVLA| 94m31s |8.9GB| 59s|29.2%| NVIDIA RTX 3090 |
> | **SpatialVLA + DTP (Ours)** | **126m20s** | **9.0GB** |**1m18s** |**37.5%** | NVIDIA RTX 3090 |
> | Nora| 41m13s|8.2GB| 26s|6.2%|NVIDIA RTX 3090 |
> | **Nora+ DTP (Ours)** | **56m6s** | **8.2GB** |**35s** |**11.5%** |NVIDIA RTX 3090 |
> | UniVLA| 36m3s |30.8GB| 23s|68.7%|NVIDIA  A100 40GB |
> | **UniVLA+ DTP (Ours)** | **38m8s** | **30.8GB** |**24s**|**74.0%** |NVIDIA  A100 40GB |
> |
>
> The efficiency results show that:
>
> 1. DTP introduces only a minor increase in inference time (typically a few seconds per task), while **memory usage remains unchanged.**
>
> 2. Despite this negligible overhead, DTP yields **consistent and meaningful gains in success rate across all models.**
>
> 3. Importantly, DTP operates **without modifying model architecture, adding parameters, or providing extra inputs or training**, it simply refines the model’s internal attention patterns.
>
> This demonstrates that DTP offers an efficient and lightweight improvement, providing stronger performance at small additional computational or memory cost.

---

> ### Author Response · Authors · 2025-11-30
>
> ###  **Response to Weaknesses 3: Sensitivity of the tolerance tau?**
>
> We thank the reviewer for raising the question about the choice and sensitivity of $τ$ in Section 4.1. This is an important point, particularly in relation to how well $τ$ generalizes across different task and environment settings.
>
> **1. Clarification on $τ$ in our experiments.**
>
> In our experiments, **we use a single $τ$ value per model across all tasks and environemnts**, rather than tuning $τ$ separately for each task. In other words, for a given VLA model, the same $τ$ is applied to all evaluated tasks, and this single setting already leads to consistent performance improvements. This is crucial for demonstrating the practicality and robustness of our approach. It shows that **for each model, one global $τ$ is sufficient to improve all tasks, without any per-task hyperparameter tuning.**
>
> Moreover, this behavior is not limited to the SIMPLER Benchmark, we observe the same pattern on the LIBERO Benchmark, where a single $τ$ per model is likewise sufficient to improve performance across multiple tasks. This makes $τ$ closer to a **model-level hyperparameter than a task-specific one.**
>
> This is one of the most importnat and meaningfully findings of our work, which shows strong robustness and practice of our method. Below shows the detailed hyperparameter used for each model, For clarity, the values below are fixed and reused unchanged across all experiments in Section 4.1.
>
> | Method |  Tolerance $τ$ | Number of Important Tokens $k$ | Guassian Smoothing $σ$ |
> |------------|--------------------------|---------------------------|------------------------|
> | **SpatialVLA + DTP** | **0.5** | **109** |**0.65** |
> | **Nora+ DTP** | **1.22** | **64** |**0.65** |
> | **UniVLA+ DTP** | **0.7** | **512** |**0.9**|
> |
>
> **2. Regarding real-world selection of $τ$.**
>
> The reviewer is also concerned about how τ would be chosen in real-world settings, where exhaustive sweeps are expensive. Our experiments suggest a practical strategy:
>
> * In practice, $τ$ can be selected with a coarse search over a small number of rollouts (e.g., on a few representative tasks), and then fixed for that model. Our experiment result indicates that one single $τ$ can work well across many tasks and different enviroments.
>
> * Thus, DTP does not require an expensive per-task sweep in the real world, rather, it only needs a one-time, model-level tuning step that can be performed in simulation or with a small budget of real-world trials.
>
> Overall, this supports the view that $τ$ is a robust and practically manageable hyperparameter, and that our method remains feasible even when policy evaluation in the real world is costly.
>
> **3. Purpose of Section 4.2 and why $τ$ varies there.**
>
> Section 4.2 serves a different purpose: instead of fixing $τ$, we sweep it over a wide range to study how success rate changes from smaller to larger $τ$ values. **This analysis is intentionally exploratory. It reveals a deeper insight that is central to our work:**
> * The $τ$-sweep exposes a common weakness in many current VLA models: **Their default attention patterns are often misaligned with task-relevant visual cues.**
> * **Even without modifying the architecture or inputs, optimizing internal visual attention patterns alone can significantly boost success rate.**
> * **There exists a sub-optimal $τ$ value where performance is maximized.**
>
> This is why Section 4.2 is not meant to imply that different tasks require different τ values, rather, it highlights the potential performance ceiling each model could reach if its visual attention were optimized appropriately.

---

> ### Author Response · Authors · 2025-11-30
> **Overall Conclusion**
>
> Across all three concerns raised by the reviewer, our additional experiments and clarifications strengthen the central claim of this work: DTP is a lightweight, robust, and broadly applicable method for improving VLA performance.
>
> * By evaluating beyond SimplerEnv and including the LIBERO Benchmark, we demonstrate that **DTP generalizes well across different environments and task suites, consistently improving performance on both.**
>
> * By evaluating both trained and zero-shot VLAs, we demonstrate that **DTP consistently benefits models regardless of how they were trained, reinforcing its generalizability.**
>
> * By providing detailed efficiency and sensitivity analyses, we demonstrate that **DTP improves success rate with minimal computation overhead, and that a single $τ$ per model is sufficient, making the method practical even in real-world settings.**
>
> * Importantly, DTP also contributes an **interpretability and diagnostic perspective**: by explicitly visualizing and correcting misaligned visual attention patterns, our method reveals why VLAs fail on certain inputs and how these failures can be mitigated. This offers a valuable tool for understanding semantic information flow within VLAs and supports future debugging, analysis, and research on attention mechanisms in embodied agents.
>
> Together, these results highlight that optimizing visual attention patterns, without altering architecture or adding inputs, can yield significant, transferable, and interpretable improvements across diverse VLA models, tasks, and environments.

---

### Author Response · Authors · 2025-11-30
**Summary of the rebuttals**

To the Area Chair,

We sincerely thank you and the reviewers for their careful evaluation of our work. During the rebuttal, we expanded evaluation to more benchmarks, demonstrated model-level generalization, and highlighted the interpretability value of our approach.

**Core contributions of our work:**

Our work makes a simple, powerful, and novel contribution to the VLA community: we reveal and correct a generic internal failure mode, misaligned visual attention by introducing DTP, an interpretability-driven, training-free, plug-and-play attention correction module, which is:

* **Model-agnostic**: works on three different VLAs with no architecture modifications.

* **Input-agnostic**: requires no new inputs or modifications.

* **Training-free**: requires no fine-tuning or re-training.

* **Consistent improvements**: across multiple benchmarks, tasks, environments and settings.

* **Interpretability value**: explicitly visualizes attention misalignment and explains failure modes.

* **Practical deployment**: needs only few hyperparameter per model, overhead is minimal.

* **Reveals untapped potential**: shows that existing VLAs can significantly improve purely through better attention allocation, without larger models or more data.

Below is a summary of how our rebuttal addresses the reviewers’ main concerns and reinforces the contribution of our method.

**In response to Reviewer xJud**, we substantially strengthened the paper with new evidence. We added full evaluations on the LIBERO Benchmark, including the difficult LIBERO-10 suite, **where DTP still improves performance, confirming strong generalization. We clarified that DTP benefits both trained and zero-shot VLAs, showing it is a model-level, not task-specific, improvement.** Efficiency analysis shows minimal overhead and that **a single $\tau$ per model works across all tasks and environments, confirming practical deployability**. Finally, **DTP also offers valuable interpretability, diagnosing and correcting misaligned visual attention, which is a key failure mode in VLAs.**

**In response to Reviewer dc8u**, this reviewer raised concerns about modest improvements, we clarified that the relative improvements are substantial, especially for weaker VLAs (e.g., +85% relative for Nora) and are achieved without any training, architectural changes, or additional inputs, purely through correcting misaligned visual attention. On efficiency, we provided full runtime and memory measurements, showing that DTP introduces only minor inference overhead while delivering meaningful success-rate improvements across all models. Finally, we clarified that the goal of DTP is not FLOP reduction but to **reveal and correct a systematic attention misalignment pervasive in current VLAs. Across all architectures, tasks, and benchmarks, we showed that fixing this issue leads to consistent performance gains, highlighting a meaningful and actionable failure mode in VLA models.**

**In response to Reviewer zxVV**, we addressed concerns on generality, hyperparameters in Section 4.1. We added evaluations on the more diverse LIBERO benchmark, showing consistent gains, demonstrating that DTP generalizes well beyond SimplerEnv. **We clarified that our evaluation covers 4,000+ rollouts across diverse environments, and that all hyperparameters (including $τ$) are fixed per model not per task, yet still provide reliable improvements across all models and benchmarks.** We improved notation clarity and explained that the Section 4.2 formulas define a principled upper bound that reveals significant untapped potential due to misaligned attention. **Lastly, we showed that DTP adds minimal inference overhead and is a training-free, plug-and-play module for correcting a fundamental attention failure mode in VLAs.**

**In response to Reviewer 3oSv**, the main concern about simulation-only evaluation, we explained that our focus is on a model-level failure mode (attention misalignment), and that we deliberately use two strong simulation benchmarks: **(1) SIMPLER, which has documented strong sim-to-real correlation, and can accurately reflect real-world policy behavior modes.** and (2) LIBERO, where DTP again consistently improves performance. We clarified that the reported gains, while sometimes modest in absolute terms, are non-trivial **given high baselines and the strict constraints of our method (no retraining, no architecture changes, no extra inputs), and that DTP’s primary contribution is diagnostic and corrective, revealing and fixing a widespread attention misallocation problem in VLAs**. We also (i) positioned our work relative to [1] and [2], explaining why they are conceptually related but not direct baselines.

We sincerely thank the Area Chair and all reviewers for the time and effort invested in evaluating our work. And we hope our responses and revisions effectively convey the significance and impact of the contributions.

Sincerely,

The Authors

---

### Meta-Review · Area_Chair_Mj4A · 2025-12-20

**Summary:**

**Paper Summary**

The paper studies DTP, a plug-and-play module that can be equipped to VLA methods to dynamically remove distracting tokens without modifying the model architecture or considering additional inputs. DTP is integrated into multiple VLAs and examined on benchmarks including SIMPLER and LIBERO, demonstrating consistent performance gains when each VLA works with DTP. The submission also provides ablations and visualization results to support its claims.


---

After reading the paper, review comments, and author responses, the AC summarizes the paper's strengths and weaknesses below.

**Strengths**
- The paper proposes an interesting and novel idea to discover unimportant regions and remove visual tokens from those areas. The proposed DTP is simple and can work with multiple VLAs.
- The paper provides extensive visualizations (and additional ablations in the discussion phase) to support its claims.
- The paper is well-written and easy to follow.

**Weaknesses**
- The evaluation can be further fulfilled:
	1. To make it clear, the AC appreciates the DTP idea of removing distracting visual tokens from unimportant regions. However, only the proposed DTP was examined in the paper. Other strategies to remove tokens should also be investigated and compared. Naive candidates may include random masking and background removal. The two methods referred to by Reviewer 3oSv are also great candidates. The AC understands there might be setting differences compared to this paper, but providing insights into which direction shows more promising results and potential would be highly appreciated.
	2. While the hyperparameter only requires model-level finetuning/searching, it still increases the burden of integrating it into new VLA models. Given the detailed hyperparameter settings in the appendix, it appears that each VLA method may require significantly different values.
	3. There is a lack of statistical information in the main experiments. Given that performance gains are incremental in some cases, the results could be perceived as biased or untrustworthy without these details.
	4. The AC understands it is challenging to provide real-world experiment results for all tasks. However, the AC agrees with the reviewers that providing real-world results would significantly strengthen the proposed method's generalization claims and potential contribution.

- The paper can be more self-contained: While the paper is well-written, the AC suggests making it more self-contained. Particularly, a complete problem statement of the robot manipulation tasks should be provided in Section 3.1, followed by the definition of the work's objective (i.e., removing distracting tokens to achieve higher performance).

**Reviewer Concerns:**

The AC appreciates the hard work the authors put into providing their response. The strengths of the proposed method are well-highlighted. While additional ablations were demonstrated, the AC still believes several concerns remain unsolved, particularly regarding
- the compared baselines (the strategies to remove visual tokens, not the integrated VLAs)
- the complexity of choosing hyperparameters
- the lack of statistical information in the main experiments.

**Reviewer Scores:**

The paper initially received scores of [4, 4, 4, 4], suggesting a consensus that the work is below the acceptance bar. Reviewers shared several concerns regarding hyperparameter searching, inference speed, and performance gain. In the author response, the authors provided additional ablations which may satisfy some of the reviewers' concerns. While it is likely that one or two reviewers will increase their evaluation scores to 6, the AC feels that the paper (even with the new evidence provided during the discussion) remains slightly below the borderline given the high standards of a venue like ICLR. Therefore, the AC recommends rejection.

---

### Decision · Program_Chairs · 2026-01-26

Reject